# Exploring the CO₂ fugacity along the east coast of South America aboard the schooner *Tara*

Léa Olivier[1,2,3], Jacqueline Boutin[3], Gilles Reverdin[3], Christopher Hunt[4], Thomas Linkowski[5], Alison Chase[6,7], Nils Haentjens[7], Pedro C. Junger[8,9*], Stéphane Pesant[10], & Douglas Vandemark[4]

[1]Alfred Wegener Institute, Helmholtz Centre for Polar and Marine Research, Bremerhaven, Germany
[2]Ludwig-Maximilians-Universität München, Munich, Germany
[3]LOCEAN-IPSL, Sorbonne Université-CNRS-IRD-MNHN, Paris, France
[4]OPAL, EOS, University of New Hampshire, Durham, New Hampshire, USA
[5]Fondation Tara Ocean
[6]Applied Physics Laboratory, University of Washington, Seattle, Washington
[7]School of Marine Sciences, University of Maine, Orono, Maine
[8]Department of Hydrobiology, Universidade Federal de São Carlos (UFSCar), São Carlos, SP 13565-905, Brazil
[9]Programa de Pós-Graduação em Ecologia e Recursos Naturais, Centro de Ciências Biológicas e da Saúde, Universidade Federal de São Carlos (UFSCar), São Carlos, SP 13565-905, Brazil
[10]European Molecular Biology Laboratory, European Bioinformatics Institute, Wellcome Genome Campus, Hinxton, Cambridge CB10 1SD, United Kingdom
*Present address: Institut de Biologie de l'École Normale Supérieure (IBENS), École Normale Supérieure, CNRS, INSERM, PSL Université Paris, Paris 75005, France.

*Correspondence to*: Léa Olivier (lea.olivier@awi.de)

**Abstract.** The air-sea $CO_2$ flux in the coastal ocean is a critical component of the global carbon budget, yet it remains poorly understood due to limited data, the many sources and sinks of carbon and their complex interactions. In August-November 2021, the *Tara* schooner collected over 14,000 km of $CO_2$ fugacity ($fCO_2$) measurements along the coast of South America, including in the Amazon River-Ocean continuum (https://doi.org/10.5281/zenodo.13790064, Olivier et al., 2024a). The Amazon River and its oceanic plume exhibit complex interactions, under a combined influence of many processes such as tides and bathymetry. Observations revealed a wide range of $fCO_2$ values, from up to up to 3000 µatm in the river to a minimum of 42 µatm downstream of the plume, where values were notably lower than atmospheric levels. South of the estuary, the $fCO_2$ of the North Brazil Current's waters (0-9°S) exceeds 400 µatm while along the Brazil Current (10-30°S), $fCO_2$ is around 400 µatm and decreases with temperature and distance from the equator. Due to its high variability in coastal environment, in the dataset salinity emerged as the primary driver of $fCO_2$ variability across this dynamic region. Despite strong variability, comparison with discrete samples of other carbonate parameters showed a mean difference of 2 µatm, within the range of uncertainties of the chemical formulas used for comparison. This dataset provides critical insights into the under-sampled region of the Brazilian coast, improving our understanding of coastal $fCO_2$ dynamics and their role in the global carbon budget.

## 1 Introduction

The global ocean is a sink that absorbs 26% of the anthropogenic carbon dioxide ($CO_2$) emitted into the atmosphere by the burning of fossil fuels and land use change (Friedlingstein et al., 2025). While the ocean participates in mitigating the effects of climate change by storing both heat and $CO_2$, it is also subject to profound changes such as ocean warming and acidification. Coastal and marginal oceans play a pivotal role in the global carbon cycle by connecting terrestrial, oceanic and atmospheric carbon reservoirs. The air-sea $CO_2$ flux varies spatially over the world's oceans, and some of the strongest gradients are found in the coastal regions (Landschützer et al., 2020). These regions present much higher temporal and spatial variability compared to the open ocean (Borges, 2005; Cai et al., 2006; Laruelle et al., 2014; Roobaert et al., 2019). Recent studies estimate that the uptake of $CO_2$ per unit area is even greater over continental shelf seas than over the open ocean due to the contribution of the arctic shelves and the impact of rivers (Chen et al., 2013; Laruelle et al., 2014; Roobaert et al., 2019). Despite the fact that coastal waters play a major role in the livelihood of humans, and are strongly affected by human activities, our understanding of these waters is strongly limited by the low number of observations (Bauer et al., 2013).

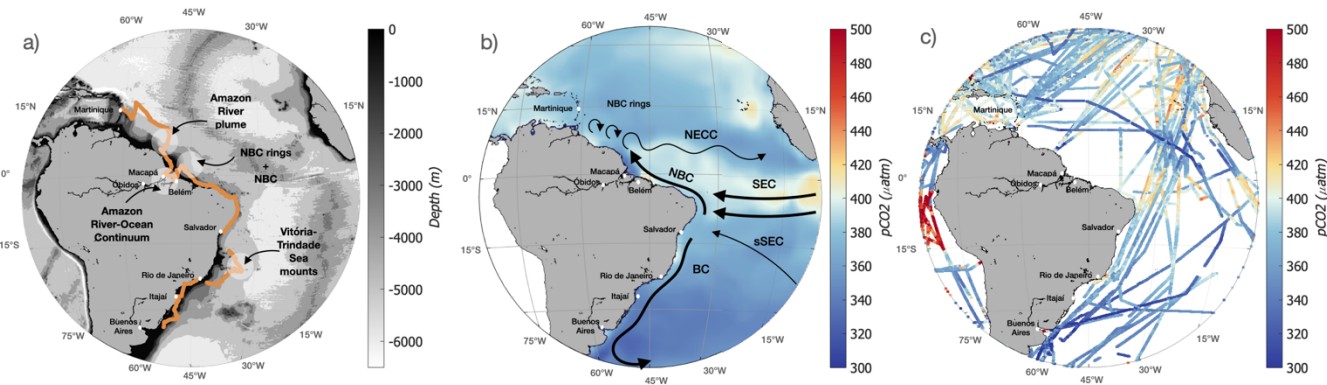

Figure 1: a) Bathymetry of the Atlantic Ocean (ETOPO2v2) and journey of the schooner *Tara* (orange line). b) 1998-2015 September climatology of the partial pressure of $CO_2$ (p$CO_2$, Landschützer et al., 2020), the black arrows represent some of the main surface geostrophic currents in boreal fall. c) p$CO_2$ along the boat trajectories in August-September-October (SOCAT, data from 1957 to 2022). NBC: North Brazil Current, NECC: North Equatorial Counter Current, SEC: South Equatorial Current, sSEC: southern branch of the SEC, BC: Brazil Current.

While the coastal and marginal seas of the mid and northern latitudes are sinks of $CO_2$ with regards to the atmosphere, the tropical coastal oceans act as sources (Cai et al., 2006; Laruelle et al., 2014; Takahashi et al., 2002). There are several reasons for this, including the reduced solubility of $CO_2$ at high temperatures, and the upwelling of deep waters rich in dissolved inorganic carbon (DIC) in the equatorial upwelling and along the coast (Andrié et al., 1986; Takahashi et al., 2002). This has also been observed on regional studies, and one region presenting a strong and heterogeneous signal is the western tropical Atlantic coastal ocean (Lefèvre et al., 2017; Lefévre et al., 2010; Olivier et al., 2022; Padin et al., 2010).

One example is the Amazon river-ocean continuum (AROC). It represents one of the greatest environmental gradients on the interface between land and ocean in the world (eg. Araujo et al., 2017). The Amazon River system discharge is unique in the

global ocean. It contributes as much freshwater as the next seven largest rivers combined, accounting for 20% of the global riverine freshwater input to the ocean (Dai and Trenberth, 2002). The resulting Amazon River plume (ARP) spreads across up to 1.3 million km² of the tropical Atlantic Ocean and creates a significant $CO_2$ sink relative to the atmosphere, primarily driven by strong biological drawdown (Cooley et al., 2007; Körtzinger, 2003; Subramaniam et al., 2008) combined with low salinities (Ibánhez et al., 2016; Lefévre et al., 2010). Opposing this, the Amazon River releases almost as much $CO_2$ into the atmosphere annually as the rainforest absorbs (Richey et al., 2002; Sawakuchi et al., 2017). The main source of $CO_2$ in the river comes from the breakdown of young organic carbon from the land by microbes (Mayorga et al., 2005; Ward et al., 2013, 2015). The lower Amazon River (from Óbidos to the river mouth) releases an amount of $CO_2$ slightly higher (0.02 Pg C yr$^{-1}$, Sawakuchi et al., 2017) than the uptake by the ARP in the Atlantic Ocean (0.014 Pg C yr$^{-1}$, Körtzinger, 2003). Sawakuchi et al. (2017) demonstrated the importance of quantifying $CO_2$ fluxes in the lower Amazon by adding the Óbidos-Macapá section to the Amazon River budget. On the other hand, oceanographic studies, carried out in particular during the ANACONDAS (in 2011, 2012 and 2013, Mu et al., 2021) and Camadas Finas III (October 2012, Araujo et al., 2017) campaigns, focused on the ARP development, maximum extension and early decay have shown the extent of $CO_2$ undersaturation in the ARP. However, the estuary, which is the link between these two systems is little known, if at all (Sawakuchi et al., 2017; Ward et al., 2017). Valerio et al., (2018) collected discrete samples for $CO_2$ partial pressure all the way to the river mouth in April 2017 but do not address the Amazon River $CO_2$ flux budget. Chen et al. (2013) studied the $CO_2$ in the world's coastal seas by evaluating the air-sea exchanges of $CO_2$ in 165 estuaries, but no data were available in the Amazon estuary, despite being arguably one with the strongest impact. Since then, Araujo et al., (2017) collected discrete DIC and total alkalinity (TA) samples at the mouth of the Pará-Tocantins River system, near the town of Belém.

The Brazilian continental shelf hosts diverse $CO_2$ flux dynamics influenced by regional oceanographic and biogeochemical processes (Kerr et al., 2016). The ARP plays a key role in air-sea $CO_2$ exchange, with strong seasonal variability driven by river discharge, biological productivity, and salinity gradients (eg. Lefévre et al., 2010; Mu et al., 2021; Olivier et al., 2024b). In the North Brazil Current (NBC) region, upwelling and mesoscale eddies contribute to $CO_2$ flux variability, modulating carbon exchange between the ocean and atmosphere (eg. Monteiro et al., 2022; Olivier et al., 2022). Further south, the Vitória-Trindade Seamount Chain interacts with regional currents (Napolitano et al., 2021), influencing nutrient transport and biological activity that can affect $CO_2$ fluxes. The Lagoa dos Patos and Guanabara Bay are important estuarine systems where terrestrial carbon inputs, tidal mixing, and anthropogenic influences create spatially and temporally variable $CO_2$ flux patterns (Cotovicz Jr et al., 2015). Along the broader Brazilian continental shelf, complex interactions between ocean circulation, biological productivity, and local conditions shape regional carbon dynamics, making in situ observations critical for understanding these fluxes.

While data gaps in the open ocean have begun to narrow, partly due to advancements such as the Argo biogeochemical float program, it is not the case for biogeochemical measurement on the shelves and continental margins. Continuous surface

fugacity of $CO_2$ ($fCO_2$) measurements carried out on ships remain the most accurate way to asses $CO_2$ fluxes and are still too sparse (Friedlingstein et al., 2025). A notable trend in recent years is the global decline in ship-based $CO_2$ observations being added to the Surface Ocean $CO_2$ Atlas (SOCAT) database (Bakker et al., 2016), particularly since 2017 (Friedlingstein et al., 2025), mainly due to reduced funding (Dong et al., 2024). Despite recent contributions documented in publicly available open-access data, the Brazilian continental margins remain notably under-sampled, with an acute lack of data during specific seasons, such as from August to November (Fig. 1c). Recently, sailboats have provided interesting opportunities to measure $CO_2$ in conditions different from traditional research vessels, as highlighted by the contribution of data from racing sailboats (Landschützer et al., 2023).

Here, we present a new $fCO_2$ data set, acquired on the research schooner *Tara*. This is the first time that a sailboat has been equipped with a $fCO_2$ equilibrator-system, which is more accurate than the membrane system used on racing yachts, but larger and more maintenance-intensive. One of the special features of the missions aboard *Tara* is the combination of physical, biogeochemical, and biological oceanography to provide comprehensive knowledge of the ocean (Bork et al., 2015; Pesant et al., 2015). Tara missions have a unique design, they are continuous for a multi-year duration, with scientists and sailors taking turns on-board. This novel dataset presents 14,000 km of $fCO_2$ measurements over 98 days between August to end of November 2021, primarily along the South American coast, and marking the first repeated sampling of the AROC. The cruise took place in a period of decreasing river outflow, following one of the largest Amazon flood events on record. Freshwater transport was strongly directed toward the Caribbean, with comparatively less Amazon-derived freshwater reaching the NECC and central Atlantic (Olivier et al., 2024b). It also includes measurements in the ARP and in different areas off Brazil: in the North Brazil Current (NBC), the Brazil Current, the Guanabara Bay (Rio de Janeiro), the Vitória-Trindade Sea mounts and the shelves of South Brazil, filling some of the gaps in the current data.

The primary objective of this study is to present the $fCO_2$ dataset acquired by *Tara*, and shed light on some of the lesser studied areas of the Amazon River estuary and plume. Section 2 provides a detailed description of the $fCO_2$ measurement system, the challenges encountered during its installation on the schooner, the solutions implemented, and the validation of the dataset. Section 3 illustrates the dataset, first encompassing the whole transect and then focusing on the case study of the river-ocean continuum. Section 4 discusses possible uses of the data and the performance of the system.

## 2 Instruments and methodology

### 2.1. Mission Microbiomes AtlantECO

For two years, the schooner *Tara* sailed 70,000 kilometers across the South Atlantic Ocean to study the ocean microbiome and its interactions with climate and pollution. The 36 m long schooner is equipped with numerous scientific equipment operated by a team of four to six scientists and six sailors consistently on board. During the first part of the Mission Microbiomes

AtlantECO, the schooner sampled the entire east coast of South America, from August to end of November 2021 (Fig. 1a). The dataset presented in this study focuses on the underway data collected during the legs 5, 6, 7, 8 and 9 of the Mission Microbiome (Table1). The dataset covers 14,000 km and stops on 25 November, as the authorization to sample in the exclusive economic zone of Uruguay was not obtained. Attempts were subsequently made to restart the system during leg 11, but these were aborted, as the conditions of the standard gas cylinders did not allow the same accuracy to be achieved.

**Table 1: Port to port description of SV *Tara*'s journey, including latitudes ranges to indicate spatial coverage.**

| Leg number | Departure | | | Arrival | | |
|---|---|---|---|---|---|---|
| | Port | Date | Latitude | Port | Date | Latitude |
| 5 | Fort-de-France (Martinique) | 18/08/2021 | 14.96 °N | Macapá (Brazil) | 9/09/2021 | 0.06°S |
| 6 | Macapá (Brazil) | 12/09/2021 | 0.06 °S | Belém (Brazil) | 17/09/2021 | 1.28 °S |
| 7 | Belém (Brazil) | 24/09/2021 | 1.28 °S | Salvador (Brazil) | 09/10/2021 | 12.41°S |
| 8 | Salvador (Brazil) | 17/10/2021 | 12.41°S | Rio de Janeiro (Brazil) | 3/11/2021 | 23.21°S |
| 9 | Rio de Janeiro (Brazil) | 11/11/2021 | 23.21°S | Santos (Brazil) | 11/11/2021 | 23.98°S |
| 9 | Santos (Brazil) | 13/11/2021 | 23.98°S | Itajaí (Brazil) | 15/11/2021 | 26.91°S |
| 9 | Itajaí (Brazil) | 19/11/2021 | 26.91°S | Buenos Aires (Argentina) | 25/11/2021 | 34.60°S |

### 2.2. Underway fCO$_2$-system

An equilibrator-based system from the university of New Hampshire (Vandemark et al., 2011) was installed on *Tara* in July 2021. It monitored continuously the near-surface ocean fCO$_2$ (Fig. 2). Currently, an equilibrator-based fCO$_2$ system is the most reliable and accurate instrument to measure the in-situ fCO$_2$ in seawater. It is able to capture the fine scale variability of oceanic

fCO₂ by responding quickly to $fCO_2$ changes in seawater. The exchange time for the water in the equilibrator is between 30 and 45 seconds, depending on flow rate (Pierrot et al., 2009). Unlike a traditional research vessel (RV), space and time for maintenance are limited on board the schooner. The continuous water line is used by 9 instruments, stored under the floor in the fore hold, and in a small laboratory of a 2.5 m². The installation of the $CO_2$ system required some compromises chosen with the help from the sailors and engineers on board, to fit with the schooner constraints and to limit the loss in measurement

accuracy (originally of less than 2 µatm). We will detail the modifications in the setup of the $fCO_2$ system, and then discuss the accuracy of the data obtained, before illustrating the large variability of the sampled area.

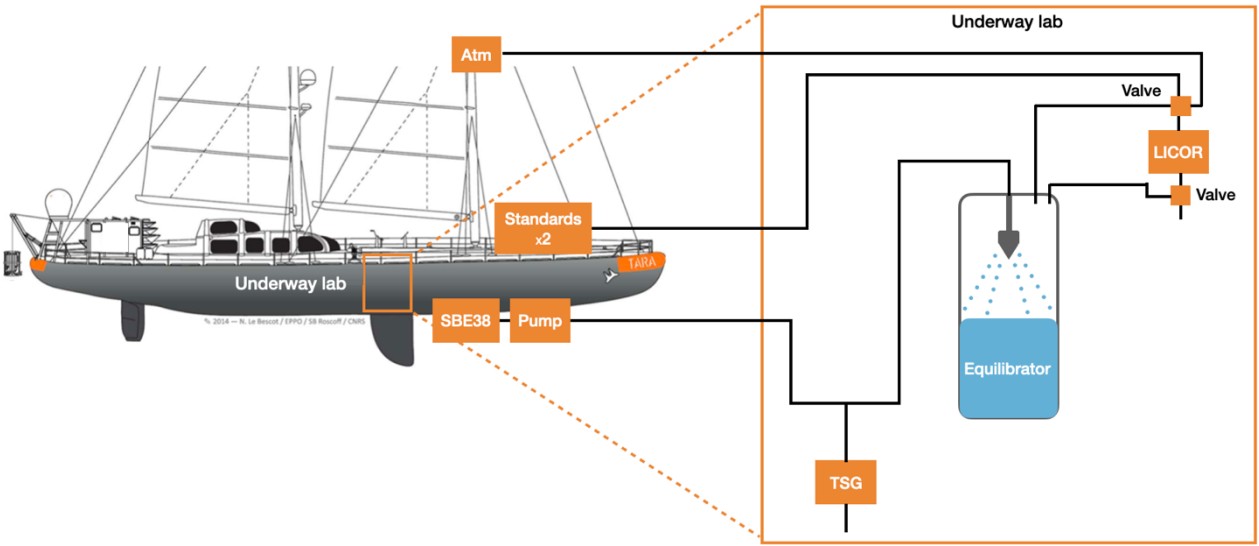

**Figure 2: Schematic of the underway laboratory and fCO₂ system onboard the schooner *Tara*, adapted from Pesant et al., (2015).**

Seawater enters through the hull at less than 1.5 m depth where a Sea Bird Electronics (SBE) 38 temperature sensor is located

for an accurate measurement of sea surface temperature (SST). It then enters a debubbler to remove most of the bubbles that can be caused by such shallow water intake, especially in rough seas, and goes through a large particles filter. The seawater circuit is then split into two to feed the many underway instruments. One branch first flows through a thermosalinograph (TSG, SBE 45) to measure temperature and salinity (temperature accuracy of $\pm$ 0.002 °C, conductivity accuracy of $\pm$ 0.0003 S m⁻¹). The other branch first flows into the equilibrator of the $fCO_2$ system after less than 5 m of tubing, at a rate of 2-6 L min⁻¹. We

make the hypothesis that at this high flow rate and short path, the temperature is similar in the equilibrator and in the TSG, as both instruments are first in their respective water circuit and at a similar distance from the split. We suspect that this choice introduces less uncertainty than directly using SST (action recommended for missing equilibrator temperature data, Pierrot et al., 2009), although we do not have the data to fully validate this hypothesis. Fortunately, the sailboat is small compared to an

open ocean RV, so the pipe length from the hull to the underway instruments is considerably smaller and the temperature

difference between the hull sensor (SBE38) and the TSG is small (always below 0.1 °C averaging 0.07°C, Figure 2).

The $fCO_2$ system uses a shower spray air–sea equilibrator of 2.5 L as described by Dickson, (2007) and used by Vandemark et al., (2011). Water is sprayed or trickled inside a chamber, creating a large surface area for rapid equilibration with the headspace air. A closed loop of air flows through the equilibrator where the air-water exchanges happen, the equilibrated air is drawn at 100 mL/min through tubing containing a Nafion selectively permeable membrane with a counterflowing stream of

dry nitrogen to remove water vapor from the sample gas stream. It is then sent to a non-dispersive infrared $CO_2$ analyzer, a LICOR LI-840A. It detects the molar fraction of $CO_2$ ($xCO_2$) in dry air by infrared detection, from which $fCO_2$ is computed following Henry's law (detailed in the annex of Pierrot et al., 2009). Ideally, the computation requires the pressure inside the equilibrator. The equilibrator was not equipped with a pressure sensor, but was designed to be at atmospheric pressure. Atmospheric pressure was measured by a Vaisala Barometer PTB100 with an accuracy of $\pm 0.3$ hPa at 20°C at the rear of the

ship. A temperature correction to the seawater $fCO_2$ data is applied based on the difference between the temperature sensor in the hull and the TSG.

Through a system of electro-valves, four circuits are operated, one for the atmospheric air, one for each of the two reference gases, and one for the air equilibrated with seawater. The atmospheric air intake is located on the first cross tree of the schooner front mast (~ 10 m). The two 20 L reference gases tanks of 0 ppm and 502.3 ppm are stored on the front deck. These values

were chosen because they effectively bracket the range of oceanic $fCO_2$ values in this highly variable environment, encompassing most of the observed data except in the river. As a result, $fCO_2$ values above 500 µatm are more uncertain and should be interpreted with caution. The number of different calibration gases was reduced from 4 (on a traditional RV) to 2 (as done on racing sailboats, Landschützer et al., 2023). The reduced use of standards results from complications to replace the gas cylinders abroad (especially during the covid period), as well as to store them onboard. It is recommended to measure

a complete set of standards every 3 hours. During the first week, to test the system, a complete set of standards and atmospheric cycle was measured for 15 minutes every hour. As the system behaved well and the drift of the LICOR was acceptable (less than 0.4 ppm over 6 hours), the measurement of standards was changed to every 6 hours (on 31/08), then to every 12 hours (on 02/09) to save the reference gases. Although not ideal, this is still more frequent than the once-a-day rate on the racing sailboats (Landschützer et al., 2023).

It is quite challenging to install such a system on a schooner, but it also presents numerous advantages, one of the most important being the shallow depth of the seawater intake. *Tara*'s seawater intake is located below the hull, at 1.5 m depth. This is shallower than on many research vessels (5 m depth on average), and better represents the actual air-sea exchanges, especially in stratified regions (Ho and Schanze, 2020). The system also has the advantage to be able to work in turbid environment. The equilibrator was cleaned at each stopover, and each time the ship exited a major river (so 7 times in total) to

avoid the buildup of mud, and the system therefore recorded data during the whole time spent in the Amazon River.

### 2.3. Using atmospheric CO₂ to validate the span value

It is customary to measure in the laboratory the value of each non-zero standard after the cruise as its value can differ from the value reported by the constructor. Unfortunately, this was not possible because after this long cruise (and some gas leakage in rough seas) the tank was empty. The value requested for the non-zero standard was 500 ppm, with a reported 507.9 ppm value by the supplier (Airgas), with an uncertainty of 2 %. The value measured in the laboratory before the cruise was 530 ppm. However, we found that choosing this value of 530 ppm results in unrealistically high atmospheric (close to 440 ppm) and oceanic (> 450 ppm) xCO₂ measurements (blue in Fig. 3). Furthermore, it is outside of the uncertainty range reported by the manufacturer.

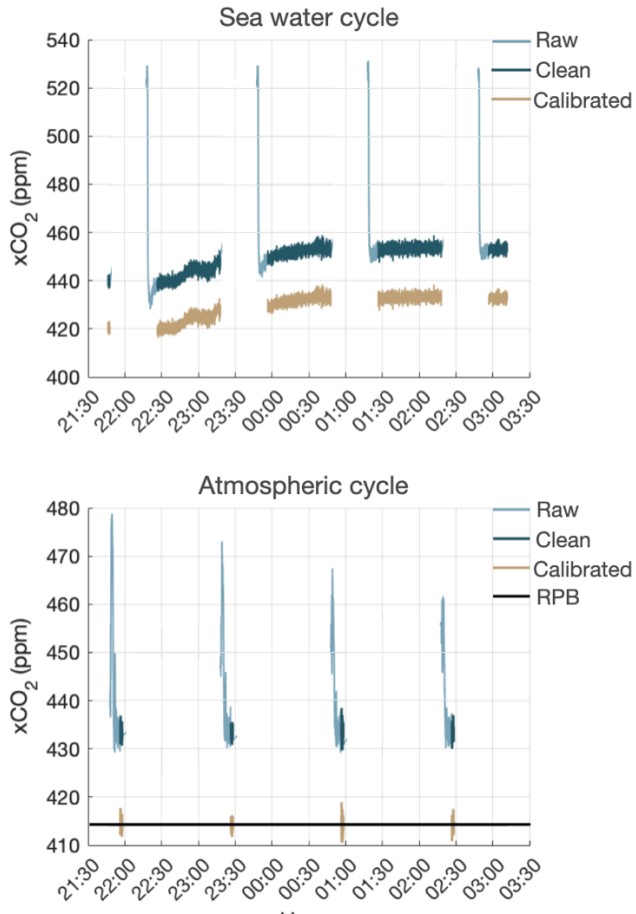

**Figure 3: Time-series of *Tara* xCO₂ extracted near Barbados (night of 18-19 August 2021) for the seawater cycle (top) and atmospheric cycle (bottom). The raw data are shown in light blue, the data cleaned for valve change pollution in dark blue and the clean and calibrated data in light brown. For the atmospheric cycle, the value measured at Ragged Point, Barbados (RPB) on 15 August 2021 (414.15 ppm) is shown in black.**

To address this calibration issue, we take advantage of the atmospheric $xCO_2$ measured on board. A few days after departure, during the night from 18 to 19 August 2021, *Tara* sailed in close vicinity to the Island of Barbados, where recurrent accurate measurements of atmospheric $xCO_2$ are taken at Ragged Point, Barbados (RPB). The $xCO_2$ measured by *Tara* calibrated using the value of 530 ppm for the span was very stable over 4 hours at 437.09 ppm. The atmospheric $xCO_2$ measured at RPB on 15 August 2021 (closest measure to *Tara*'s passage) was 414.245 ppm. In order for the *Tara* $xCO_2$ data near Barbados to match

the ones at RPB, the span value should be 502.3 ppm, which is within the uncertainty range provided by Airgas. This span value was then used to calibrate the entire dataset (light brown in Fig. 3). This approach assumes that the atmospheric $CO_2$ near Barbados is representative of the value on *Tara*. During the time *Tara* was near the island, winds were moderate and blowing from the sea (not shown), so the atmospheric $xCO_2$ at RPB is not expected to vary much from day to day (~ 0.5 ppm). In the worst case, an error of 1 ppm in the calibration value would lead to a $\pm$ 0.84 ppm averaged difference over the dataset.

*Tara* crossed highly variable regions during its voyage, supporting our confidence that the uncertainty on the dataset associated with this span value has limited influence on the overall results, but should nevertheless be take into account when analyzing the dataset.

## 2.3. Validation of the dataset

Samples of TA and DIC were taken at each station. Out of a total of 78 samples, 17 are from the surface (Metzl et al., 2024). Samples were drawn from the rosette into 0.5 L borosilicate glass bottles, ensuring minimal air contamination, and immediately poisoned with 400 µL of mercuric chloride ($HgCl_2$) to prevent biological alteration. TA was measured using open-cell titration with a hydrochloric acid titrant, while DIC was analyzed using acidification followed by $CO_2$ extraction and detection via infrared or coulometric methods. Quality control was ensured through calibration with certified reference materials to maintain

an accuracy of $\pm$4 µmol kg$^{-1}$ (Metzl et al., 2024). Their salinity ranges on a salinity scale from 0 inside the Amazon River to 37.3 in the North Brazil Current. The $fCO_2$ is computed from the near-surface ocean DIC and TA using the CO2SYS v3.1 software (Sharp et al., 2020) to compare with the continuous $fCO_2$ measurements (Fig. 4). The dissociation constants were taken as the one of Mehrbach refitted by Dickson and Millero (Dickson and Millero, 1987; Mehrbach et al., 1973) and nutrients were neglected. The dissociation constants used are the same as those in Lefevre et al. (2010) to ensure consistency for

comparison in Section 4. However, we also tested several other sets of constants for additional analysis, detailed below. It is worth noting that DIC and TA is not the most accurate pair to determine the $fCO_2$, and it can lead to a probable error of 5.7 µatm (Millero et al., 1995).

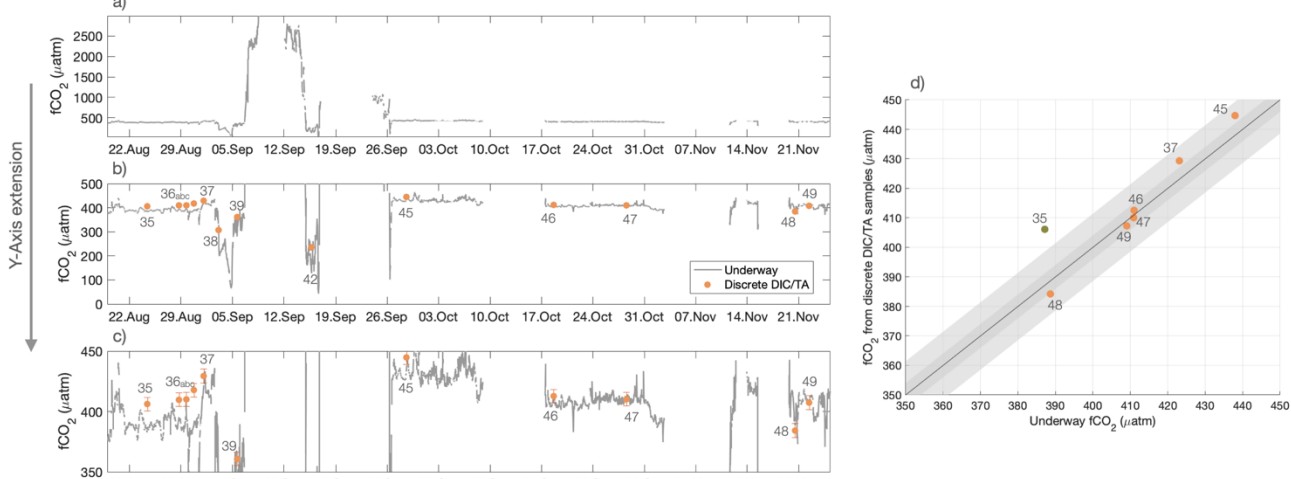

**Figure 4: Time-series of the surface fCO₂ from 18/08/2021 to 25/11/2021, for the full range of values (a) for only oceanic values (b) for values between 350 and 450 µatm (c). The dots indicate the fCO₂ inferred from the DIC/TA water samples for stations 35 to 49, with error bars of 5.7 µatm to represent the uncertainty of the chemical formulas. Scatter plot of the underway fCO₂ and the fCO₂ inferred from the DIC/TA samples, for fCO₂ values ranging between 350 and 450 µatm. The green dot indicates a salinity difference between the CTD sensor and the sample from bottle of more than 0.5. The fCO₂ system was measuring the standards and not seawater during stations 36abc and 39, so these stations are not represented in (d).**

Considering the compromise on accuracy that had to be made to be able to sample on the schooner, the continuous fCO₂ compares well to the one computed from the samples, especially after 26 September. Before that, the DIC/TA samples from the rosette during CTD casts were often taken in the presence of salinity-stratification near the surface in the ARP. The depth actually sampled for these samples is likely to be a bit deeper (by a couple of meters) than the depth of the in-line TSG (sampling depth at 1.5 m), which could explain why the fCO₂ measured underway is lower than the one inferred from DIC/TA in the ARP. For station 35, the difference of salinity of 0.9 between the salinity sample from the Niskin bottle and the CTD sensor is an indication of the high temporal and spatial variability of the area. Over the whole time-series no constant bias is identified, and the mean difference after 26 September is of 2.02 µatm (standard deviation of the difference ($STD_{diff}$) =7.4 µatm), and drops to 0.97 µatm ($STD_{diff}$ = 0.5 µatm) after 10 October. These results vary but remain in the same order of magnitude when changing the dissociation constants. Using constants from Lueker et al., (2000) leads to similar results (mean difference of 1.2 µatm), the largest differences are obtained using the constants from Waters et al., (2014), that are designed for a large salinity range (0-50). It improves the comparison for low salinities (before September 26) but gives slightly larger differences for high salinities (mean difference after 26 September of 0.5 µatm and 3.4 µatm after 10 October). Overall, the mean difference remains around 2 µatm, providing a reasonable estimate of the dataset's uncertainty. In the river, where fCO₂ values fall outside the range of the standard gas used and no discrete samples are available for direct comparison, the uncertainty is likely higher. However, the values obtained align with expected ranges for this part of the river, based on discrete samples collected in April 2017 by Valerio et al. (2018), despite differences in season and year.

As no simultaneous dataset can be used to cross-quality check the data, the agreement tendency between $fCO_2$ estimated from 6 samples and the continuous $fCO_2$ measurements is important, and is used here to validate the data. However, the number of samples—particularly in the open ocean—is very limited relative to the distance covered, which limits the statistical robustness
of the validation of both the dataset and more importantly of the calibration approach. Users should therefore interpret the data, especially in coastal and river-influenced regions, with appropriate caution.

## 2.5. Reported data

Following the recommendations of Pierrot et al. (2009) and of SOCAT, the dataset provides for each location and time step the measured data: molar fraction of $CO_2$ in the equilibrator ($xCO2_{eq}$), sea surface salinity (SSS), temperatures (SST and $T_{eq}$),
and pressure ($P_{atm}$), the calculated variables ($pCO_{2sw}$, $fCO_{2sw}$), averaged over one minute. The atmospheric $fCO_2$ is not included as the atmospheric $xCO_2$ was used as a standard and for validation of the dataset. The dataset will be submitted to the 2025 SOCAT version, with probably a flag C, as only one non-zero reference gas is used to calibrate the measured $xCO_2$. In the meantime, the data are available in the following public repository: https://zenodo.org/records/13790064 (Olivier et al., 2024a). In the dataset, ancillary data are added (wind speed at 10 m, bottom depth from ETOPO2v2) to achieve a more detailed
interpretation of the data. The wind speed was measured by a Gill anemometer at the top of the mast (27 m), and then adjusted to 10 m using a logarithmic relationship (Tennekes, 1973) and can be used for the calculation of the $CO_2$ flux. This dataset addresses the overall lack of data identified by SOCAT, by covering diverse environmental gradients with a high-resolution sampling. The use of the schooner highlights the potential of non-traditional platforms for collecting high-quality data in challenging environments, complementing traditional research vessels.


# 3 Overview of the data

## 3.1. From the Caribbean to Uruguay

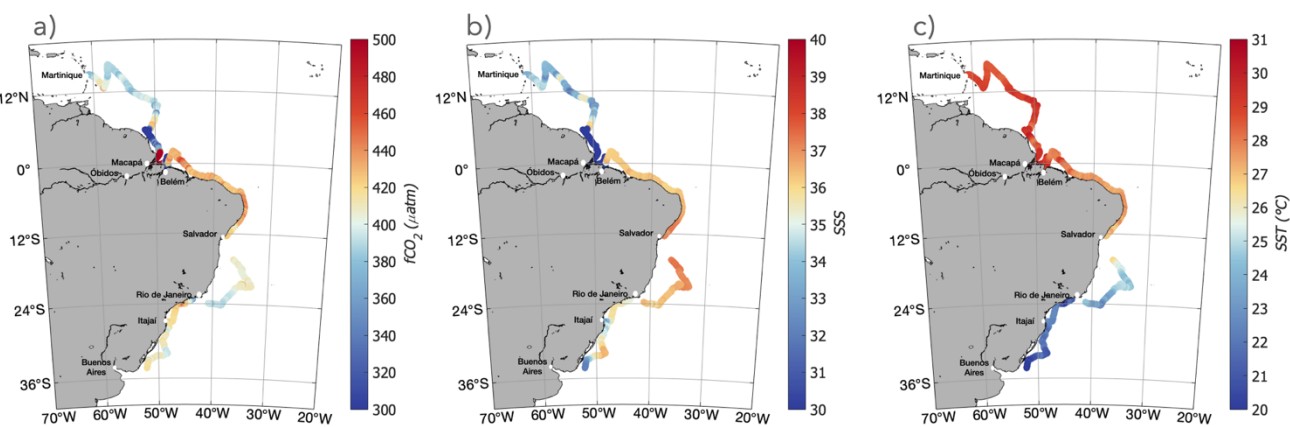

**Figure 5: Along track a) CO$_2$ fugacity (complete range of values shown in Figure 4), b) sea surface salinity and c) sea surface temperature.**

After leaving Martinique Island on 18 August 2021, *Tara* sampled the Northwestern tropical Atlantic. Surface waters exhibited strong spatial variability, with temperatures and salinities changing from 27.5 °C to 30.5°C and from 31.5 to 35.5. It induced a variability of the surface fCO$_2$, that ranged from 370 to 420 µatm (Fig. 5 and 6).

The schooner then crossed the salty (36) water of the NBC retroflection, before sampling the river plume that had been recently transported from the Amazon estuary. Around this period of time, the ARP was located almost entirely on the shelf as salinities lower than 30 were observed at depths shallower than 100 m (Fig. 6). The ARP water is drastically different from the one of the NBC retroflection, and the two water masses are separated by strong horizontal fronts. On 3 September 2021 *Tara* crossed a front of 14.2 in salinity between 00h30 and 5h00. This first strong front was followed by several others, on 4 September between 00h and 20h (loss of 14 salinity unit), between 4 September 20h and 5 September 9h (increase of 17 salinity unit), and finally on the 6 September between 10h and 23h the salinity dropped from 24.2 to 0 as the schooner reached the Amazon River. These sharp salinity fronts are associated with variations of temperature (variability of 2-3°C) and mainly fCO$_2$. In the ARP, the fCO$_2$ variations follow the ones in salinity. The fCO$_2$ of the ARP is extremely low, as for a salinity of 11 on 4 September a fCO$_2$ of 65 µatm is observed (Fig. 6). The salinity and fCO$_2$ increase on 5 September are associated with a decrease of SST, which could suggest an event of vertical mixing or local upwelling. This event generated fCO$_2$ fluctuations, and then as the schooner approaches the river and the salinity decreases, the conditions are switching from marine to riverine and fCO$_2$ rapidly increases. In the Amazon River, fCO$_2$ is very high, reaching 3000 µatm in Macapá.

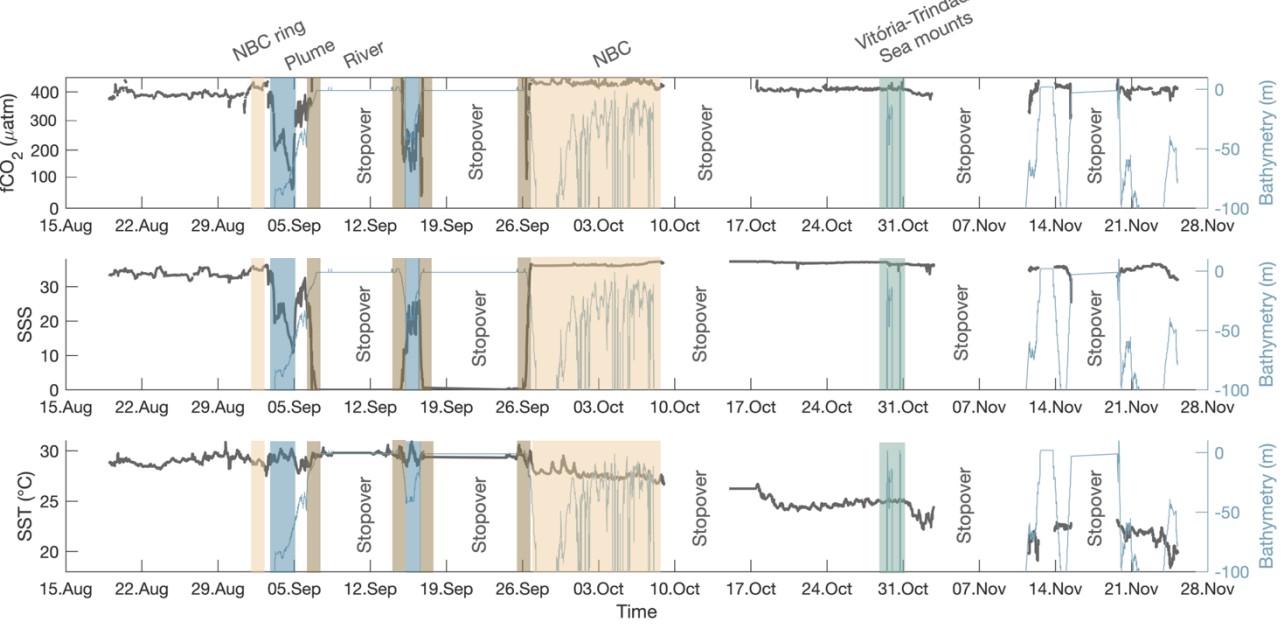

**Figure 6: Time-series of fCO₂, sea surface salinity and sea surface temperature. The light blue line on each panel represents the along track ETOPO2v2 bathymetry. The shaded patches show areas of interest identified on Fig.1. NBC: North Brazil Current.**


On 12 September, the schooner left the Amazon River and sampled the Amazon and Pará Rivers plume before entering the Pará River to join Belém. The lowest $fCO_2$ of the time series was observed in the Amazon/ Pará River plume, with a $fCO_2$ of 42.8 µatm offshore of the Pará River. After the stopover in Belém, the ship sampled the waters of the NBC. The temperature decreased and showed a variability on the order of one degree Celsius. The NBC waters stand out by their high salinity (around

37) and high $fCO_2$ (~ 420 µatm), and strongly contrast with the river plume waters. From 9 October to 18 October the ship stopped in Salvador, and then sailed to Rio de Janeiro, with a particular focus on the Vitória-Trindade Seamounts, a biodiversity hotspot (Pinheiro et al., 2015) amidst the South Atlantic Subtropical Gyre, one of the most oligotrophic zones of the global ocean (Morel et al., 2010). The temperature was colder (24°C after Salvador compared to 26/27°C before), and its variability is closely associated with the one of $fCO_2$ (the decrease in temperature is associated with a decrease in $fCO_2$ close to the rate

of 4.23 %/°C given by Takahashi et al., 1993). This indicates a switch from a $fCO_2$ variability dominated by salinity and primary production to a $fCO_2$ variability dominated by a temperature solubility effect. Around the Vitória-Trindade Seamounts (28 October to 1 November) we observe a strong variability of surface salinity and $CO_2$, correlated to the shallower bathymetry, which could be driven by upwelling turbulent mixing (Mashayek et al., 2024; Napolitano et al., 2021).

The ship called in Rio de Janeiro from 2 November to 11 November and then sailed to Itajaí with a call in Santos. This part of

the journey is very coastal, with bottom depth almost always above 100 m. It shows strong $fCO_2$ variability, with low values associated to the lower salinities (34.8) close to Santos. After 19 November, the temperature decreased as the ship sailed

southward reaching a minimum of 18.4 °C, associated to a small drop in fCO₂. Salinity also decreased to 31 as bottom depths get shallower than 50 m, possibly an early signal from the Rio de la Plata plume or/and a signal from the Lagoa dos Patos.

### 3.2 The Amazon river-ocean continuum

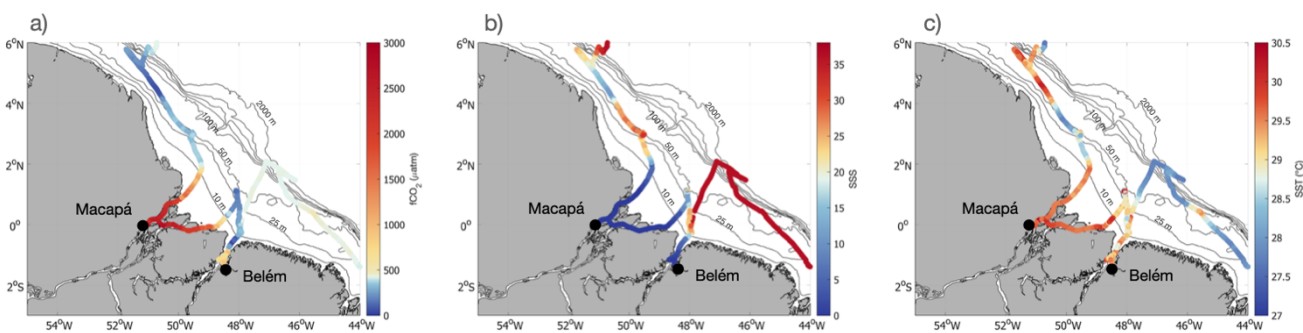

**Figure 7: Along track fCO₂ (a), sea surface salinity (b) and sea surface temperature (c) in the Amazon region. Bathymetry contours are represented in black, from 10 m to 2000 m.**

The largest variations of fCO₂ are observed in the AROC. The strongest gradient, reaching 3000 µatm, is observed at the transition between the marine and riverine waters. The signature of the ARP in itself is important, with a variation of fCO₂ of up to 340 µatm (Fig. 6). The minimum observed fCO₂ in the Amazon River plume is of 65 µatm (4.5°N/50.77°W), whereas outside of the plume, in the NBC, the fCO₂ is around 420 µatm. Then, on the Amazon shelf, fCO₂ progressively gets stronger as *Tara* went southward towards the Amazon estuary, with the change in regime intensifying around the 30 m bathymetry line, when the ARP switches from a sink to a very strong source (fCO₂ > 2000 µatm, Fig. 7). The AROC is sampled twice, and so is the Pará river-ocean continuum (Fig. 6, 7).

For the four crossings of the river-ocean continuum (two in the Amazon River, two in the Pará River), different fCO₂/SSS relationships (Fig. 8) and different relationships to bathymetry (Fig. 9) are observed. From the NBC to the core of the ARP (6°N to 4°N, in dark blue on Fig. 8a, b), the fCO₂/SSS measurements follow well the relationship reported in Lefèvre et al., (2010, then NL). However, when salinity and fCO₂ increase locally from 4°N to 2°N (light blue and brown on Fig.8), they move away from the NL linear relationship. This is even more pronounced closer to the Amazon River, where the salinity decreases (from 25 to 0) and bottom depth is shallower than 20 m (Fig. 9). After a slow decrease for salinity ranging from 25 to 12, fCO₂ sharply increases in a non-linear fashion as the bathymetry gets shallower than 20 m (salinity ranging from 12 to 5). For depth shallower than 10 m and salinities below 5, the fCO₂ is already greater than 1000 µatm and shows the largest variability (on the order of 500 µatm) before the ship enters the pure riverine waters (0 salinity, depth of ~ 2 m, Fig. 8b, 9b).

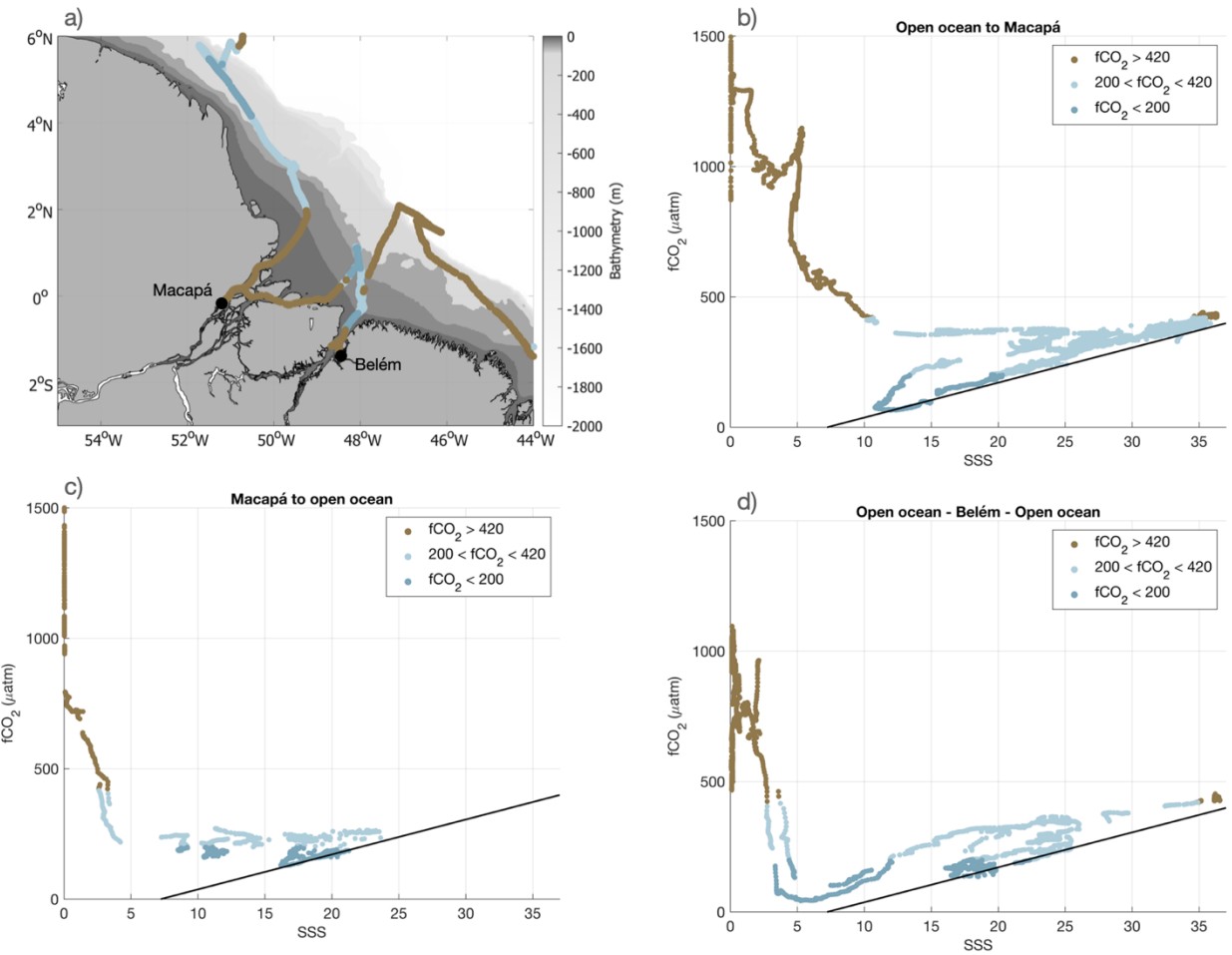

**Figure 8: a) Map of the Amazon region, with bathymetry contours. Each region is defined on Figure 9. The track is colored based on the fCO₂ values to highlight three regimes (fCO₂ >= 420 μatm (brown), 200μatm < fCO₂ < 420μatm (light blue) and fCO₂ <= 200 μatm (dark blue)). fCO₂-SSS diagram for the entrance in the Amazon River (b) for the exit of the Amazon River (c) and for the sampling of the Pará River (d). The black line on b) c) d) is the fCO₂/SSS relationship from Lefèvre et al., (2010).**

The schooner leaves the Amazon River through a different branch than on entry (Fig. 8a). The decrease in fCO₂ from 1000 μatm to 300 μatm is more linear with respect to salinity, and the source-sink transition occurs at a lower salinity than on entry (3 instead of 12, Fig. 8c). The points in the 5 to 25 salinity range show great variability, and only those with the lowest fCO₂ and salinities between 15 and 20 follow the NL relationship. The variation of salinity with bathymetry is not the same as in the way in, where the salinity stays lower than 15 for depths between 20 and 40 m.

The variation of $fCO_2$ along the way in the Pará River is also different from the way out, but with less variability than for the Amazon River (Fig. 8d). The sink-source transition occurs at salinities of 2.7 and 3.7 respectively and for very shallow depth
(< 5 m, Fig. 9a). The minimum $fCO_2$ reached before the Pará River is 42.7 µatm for a salinity of 5, whereas on the way out it is 101.6 µatm for a salinity of 7. This is likely due to the Pará River plume being advected northwestward along the shelf and mixing with the ARP on its northern side. On the southern side, it mixes with the carbon-rich waters of the NBC.

The transition from a source to a sink thus presents large variability. It doesn't happen at a consistent salinity or bottom depth,
highlighting the role of other parameters driving the $fCO_2$ variability in the region.

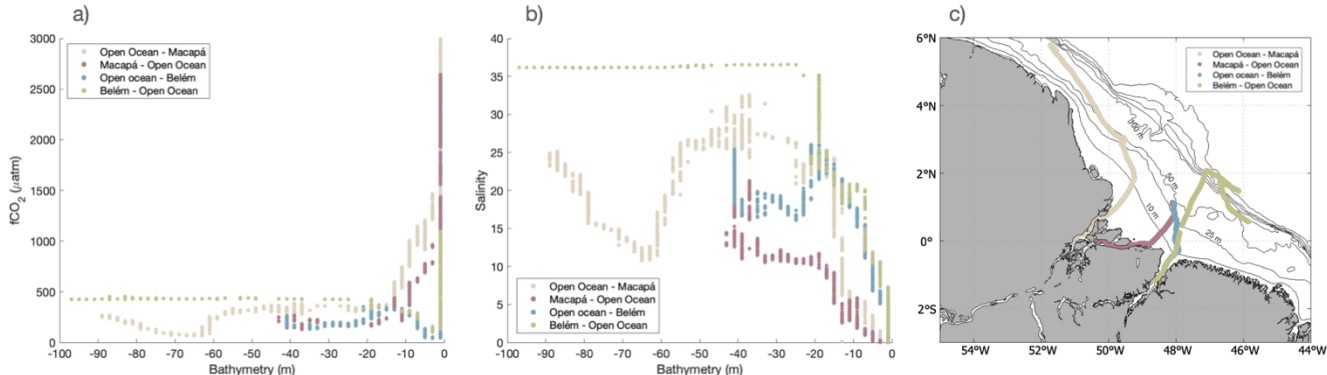

**Figure 9: a) $fCO_2$-bathymetry and b) SSS-bathymetry of the different river-ocean continuum crossings for depths shallower than 95 m. The bathymetry is ETOPO2v2 colocalized along the ship track.**

**4 Discussion**

**4.1 Main drivers of $fCO_2$ variability**

       **4.1.1 Salinity**

In the equatorial band (15°N-15°S) of the western Atlantic, the $fCO_2$ variability in the data follows well the strong surface salinity variability (Fig. 6). There are two reasons for that. First, in this region, the surface temperature is very warm (SST>27°C), with relatively small variability (the STD of the SST in the dataset is of 0.8°C). The solubility effect associated
to an increase in temperature of 0.8 °C would be an increase of the $fCO_2$ by 13.6 µatm (Takahashi et al. 1993, 4.23%/°C). While this is non-negligeable, it is small compared to the observed STD of 109 µatm observed in the *Tara* dataset between 15°N and 15°S (excluding waters with SSS<=1).
Second, the Amazon River flows into the tropical Atlantic and forms huge salinity gradients (STD of the SSS in the *Tara* dataset between 15°N and 15°S is of 7.5 (for SSS>1)). These gradients indicate the river's influence on the open ocean, and
thus also the changes in biogeochemical properties. At first order, the gradient in salinity is indicative the gradient in alkalinity

and DIC (linked to biological activity) associated to the river plume and therefore fCO$_2$. This explains the robustness of empirical linear relationships between salinity and fCO$_2$ in the ARP, such as the one of Lefèvre et al., (2010), presented in part 3.2.

### 4.1.2 Temperature

South of 15°S, the situation is different, with larger temperature changes. As the schooner sails poleward, the temperature decreases with no more influence of the Amazon River system. At first order, the fCO$_2$ variations follow the one expected from the solubility effect. The SST decreases of 8°C, and the change in fCO$_2$, with a maximum of 442.8 µatm and a minimum of 309.2 µatm, is coherent with a drop of 8°C in temperature (expected drop in fCO$_2$ of 136 µatm following a 4.23 %/°C effect; Takahashi et al., 1993).

Nevertheless, while the large-scale variability of the fCO$_2$ reflects the latitudinal temperature gradient (Landschützer et al., 2013), at smaller scales the variability of salinity is also important and different water masses are sampled. Notably, south of Rio de Janeiro, the schooner sails on the shelf, with a bathymetry often shallower than 100 m (Abril et al., 2022). Other river discharges reach the south Atlantic, such as the one of the Rio de la Plata (Marta-Almeida et al., 2021). These waters spread on the shelf and generate variability in salinity, suspended sediments and biological activity (Marta-Almeida et al., 2021; Piola

et al., 2005).

### 4.2 The sink-source transition in the river-ocean continuum

The multiple crossings of the river-ocean continuum show a great fCO$_2$ variability in the Amazon River/ Pará River estuaries and resulting plumes on the Amazon shelf. As a continuous feature, this environment can extend over 500 km along-shelf and

200 km across-shelf (Curtin and Legeckis, 1986). Within this region, the transports of fresh water, sediment, nutrients and biomass are determined by energetic processes occurring on semidiurnal, several-day, several-week and seasonal time scales (Curtin, 1986; Geyer et al., 1991).

In the regions close to the river estuary, the fCO$_2$ changes are no longer primarily associated with the changes in salinity. As

salinity decreases, the switch from a decreasing fCO$_2$ (representative of the plume) to increasing fCO$_2$ (representative of the river) does not happen at the same salinity for each crossing. Moreover, the linear relationship between salinity and fCO$_2$ does not hold close to the river. It is thus likely that the salinity gradient no longer mirrors the gradient in DIC or in TA. The strong sediment load of the Amazon River prevents the light penetration in the water column and the development of photosynthetic organisms (DeMaster et al., 1986; Gagne-Maynard et al., 2017). The source to sink transition is mainly driven by the switch

from a respiration dominated system to a photosynthetic one (Gagne-Maynard et al., 2017; Mu et al., 2021). Several factors impact the suspension of sediments in the water column and the development of phytoplankton, such as the bathymetry, winds, intensity of the outflow (that can be influenced by large-scale climatic modes) and the tides (Gomes et al., 2021). Indeed, tides

and tidal currents are one of the dominant factors of the variability of the Amazon estuary with tidal currents ranging from 0.5 to 2.0 m s$^{-1}$ (Geyer et al., 1991; Ruault et al., 2020).

The relationship between suspended sediments and bathymetry allowed the identification of 4 zones of interactions by Curtin & Legeckis (1986). They match well with the $fCO_2$ measurements. For the Amazon River, the zone of highest suspended sediments concentration (SSC) is located between the isobaths 4 m and 11 m, matching the strong increase of the $fCO_2$ observed for the two crossings of the AROC. The zone of lowest SSC found by Curtin and Legeckis, (1986) and DeMaster et al., (1986) is between the isobaths 10 m and 20 m, which it is also the region where we observe the transition from a source to

a sink of $CO_2$. For the Pará River, this region extends directly to the mouth, for depth shallower than 5 m. It also matches the observations, and the differences between the two rivers. Indeed, while for the Amazon River, the transition from a sink to a source happens between 10 and 20-m depth, it happens at much shallower depths for the Pará River (below 10 m). Their "River Zone", for depth below 5 m, indeed corresponds to a salinity of 0. Nevertheless, we observe important variability of $fCO_2$ even if the salinity does not change anymore. This region was not investigated by these studies that focused further offshore of the

mouth. For the region, the bathymetry used here is not adapted anymore, and a specific Amazon estuary bathymetry should be used for further studies (such as Fassoni-Andrade et al., 2021). Therefore, combining the $fCO_2$ dataset and the numerous optical measurements also conducted underway onboard the *Tara* Mission Microbiome could lead to a better understanding of the AROC system. Moreover, linking this continuous dataset to the discrete imaging and genetic samples from Mission Microbiome's stations, conducted in the different zones of the system, will also bring light on the biological communities

responsible for the strong $CO_2$ source or sink observed.

### 4.3 Limitations of the dataset

Onboard *Tara*, there is rarely a trained scientist to take care of an equilibrator $fCO_2$ system. The system therefore has to run almost autonomously, and is monitored from land when someone trained is not onboard. Limited space meant that only two standards were used, and they were stored outside on the foredeck. The deck is subject to spray, waves and wave-related

impacts. This increases the strain on the system, and the possibility of failure, in particularly during bad weather. The $fCO_2$ system operations were finally terminated due to several leakages that happened during the strong sea state encountered in the Southern Ocean. For this mission, *Tara* sampled mainly coastal environments, where $CO_2$ is highly variable and little known. There are very little previously acquired data in the region that can be used for comparison. And even where there are other data, the variability in the coastal ocean is such that it might not be comparable. Some surface samples of DIC and TA were

collected by scientists on board, which were essential to validate the $fCO_2$ measured. The mean difference of 2 μatm and STD$_{diff}$ of 7.4 μatm (going down to 0.5 μatm in the less variable environment) give an estimate of the uncertainty that support the validity of the dataset. It is nevertheless necessary to note that the relationship to compute $fCO_2$ from DIC and TA also has an uncertainty of 5.4 μatm, and it would be more accurate to cross-compare with $fCO_2$ measurements conducted in the same region at the same time as recommended by SOCAT. This shows the limitation of autonomous $fCO_2$ systems that cannot be

checked regularly, and especially the ones on small boats that are more fragile due to the rougher conditions than on a large research or container ship.

**5 Conclusion**

For the first time, a schooner equipped with a $fCO_2$ equilibrator system measured $fCO_2$ along the eastern coasts of South
America. This high temporal resolution dataset includes $fCO_2$ measurements every minute over 14,000 km of sailing. From the Caribbean to Argentina, this dataset of 65,000 measurements spanning over almost 4 month (from August to end of November 2021, for a total of 45 days and 8 hours of valid $fCO_2$ data) shows large $fCO_2$ variability. In particular, it sampled the Amazon River plume, the Amazon and Pará River estuaries, the North Brazil Current, the Brazil Current, the Vitória-Trindade Sea mounts (local hotspot of biodiversity), and the shelves of southern Brazil.


In August-September 2021, the Amazon-Pará plume is highly undersaturated with $CO_2$, in line with the many regional studies on the Amazon River plume (Ibánhez et al., 2015; Körtzinger, 2003; Lefévre et al., 2010; Mu et al., 2021). This dataset provides data closer to the river than in some of these earlier studies, sampling the core of the plume. Further from the mouth, $fCO_2$ reaches extreme low values, between 40 and 60 µatm, which had never been observed before. It is possible to measure such
low values because for the first time a ship equipped with a $fCO_2$ system is sampling the river-ocean continuum, and pumping water at a very shallow depth. When salinity continues to drop (S<8), a sink-source transition occurs, and $fCO_2$ rises rapidly. The influence of the river becomes dominant, and $fCO_2$ reaches 3000 µatm in the river. The river-ocean continuum has been crossed four times, and each time showed different properties. This system is highly dynamic and needs to be studied further in depth to infer the global role of the Amazon system in the global carbon budget.


Equipping a sailboat with a $fCO_2$ equilibrator system is a challenge, but one that has been met by the schooner *Tara*. The dataset is very valuable for global and regional studies, providing data in the data-poor region of the coastal regions of the South Atlantic Ocean. It is particularly helpful for $fCO_2$ mapping products, which assimilate all data collected to produce global monthly and climatological $fCO_2$ maps from neural network reconstruction (Chau et al., 2024; Denvil-Sommer et al.,
2019; Landschützer et al., 2016, 2020; Laruelle et al., 2017). It is also useful for process studies, such as the river-ocean continuum (Sawakuchi et al., 2017), offshore ARP (Olivier et al., 2024b), and the coastal currents of the South American coast. The difficulty in validating the dataset shows just how little is known about coastal regions and how dynamic they are. The limited number of observations could be due to the complicated access some of these regions (distance from major port), limited funding and to the difficulty to obtain sampling permits. Collecting more $fCO_2$ data in under-sampled regions, such as
the southern hemisphere oceans, the Southern Ocean, coastal regions and estuaries, is very important to improve our knowledge of the global carbon cycle (Roobaert et al., 2019).

**Data availability**

The dataset is available in the following public repository: https://doi.org/10.5281/zenodo.13790064 (Olivier et al., 2024a),
with the DOI 10.5281/zenodo.13790064. It is also submitted to the SOCAT version 2025.

**Author contribution**

LO, JB and GR conceptualized the project. LO, TL, NH and AC collected the data and LO and CH curated the data. CH and DV designed and provided the instrument to collect the dataset. SP managed and coordinated the project on land and on board for the mission. LO prepared the manuscript with contributions from all co-authors.

**Competing interests**

The authors declare that they have no conflict of interest.

**Acknowledgements**

We wish to thank the Tara Ocean Foundation, the SV *Tara* crew and all those who participate in Mission Microbiomes AtlantECO and adopt its Data Sharing & Publication Best Practices (https://zenodo.org/communities/mission-microbiomes-atlanteco/). In particular, we would like to thank chief engineer Léo Boulon for all his help with the installation of the system as well as Martin Hertau, Nicolas Bin and Samuel Audrain for the maintenance. Clémentine Moulin and Aliénor Bourdais, thank you for the help with the logistics, the shipping and the coordination. We warmly thank the SNAPOCO$_2$ and Jonathan Fin for the precious accurate analysis of the DIC/TA samples. We are keen to thank the commitment of the following institutions for their financial and scientific support that made Mission Microbiomes AtlantECO possible: Stazione Zoologica Anton Dohrn, European Bioinformatics Institute (EMBL-EBI), Centre national de la recherche scientifique (CNRS), Centre National de Séquençage (CNS, Genoscope), agnès b., BIC, Capgemini Engineering, Fondation Groupe EDF, Compagnie Nationale du Rhône, L'Oréal, Biotherm, Région Bretagne, Lorient Agglomération, Billerudkorsnas, Havas Paris, Fondation Rothschild, Office Français de la Biodiversité, AmerisourceBergen, Philgood Foundation, UNESCO-IOC, Etienne Bourgois.

**Financial support**

This publication has received funding from the European Union's Horizon 2020 research and innovation programme under grant agreement No 862923 (project AtlantECO). This output reflects only the author's view and the European Union cannot be held responsible for any use that may be made of the information contained therein. This work was supported by the Initiative and Networking Fund of the Helmholtz Association (Grant Number: VH-NG-19-33). P.C.J. was supported by Fundação de Amparo à Pesquisa do Estado de São Paulo (FAPESP; PhD grant no. 2017/26786-1) and by FAI/UFSCar (ProEx no. 3213/2020-83) through the European Union—H2020 project AtlantECO (award no. 862923).

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
