# Peer review of "Exploring the CO2 fugacity along the east coast of South America aboard the schooner *Tara"

_Earth System Science Data, 2024_

## Referee Comment (RC1)

**Review for the manuscript: "Exploring the CO$_2$ Fugacity along the East Coast of South America aboard the Schooner Tara" by Olivier et al.**

**Time devoted to this review**: 12 hours.

**Reviewer**:
Marcos Fontela (IIM-CSIC)

This manuscript presents an innovative methodology and a timely study, leveraging the capabilities of the *Tara* schooner to monitor CO$_2$ fugacity (fCO$_2$) in a key region for the global carbon cycle. The work demonstrates the potential of such expeditions for regional carbon dynamics studies and contributes valuable observations to the limited dataset from this area. Moreover, this represents an excellent initiative to expand the scope of the *Tara* schooner's activities—traditionally centered on marine biology and ecology—toward geoscience observations, as exemplified by the dataset presented in this manuscript. The manuscript is well-written, well-structured, and includes high-quality figures. The article is worthy of publication and appropriate to support the publication of a data set, but the dataset itself exhibits significant shortcomings that must be first addressed. Important modifications are required to ensure the dataset meets the standards of openness, metadata completeness, and long-term usability. I recommend a major revision, primarily to address issues with the data product, as outlined below. Additionally, there are inconsistencies in the framing and presentation that detract from the overall clarity and impact of the work, but they should be easily ameliorated.

The current form of the data product significantly weakens the manuscript's suitability for publication in ESSD, a journal known for its rigorous standards in data quality and accessibility. The dataset lacks metadata, the data file itself it has an untraceable naming ("*CO2Tara.xlsx*" -sic-), fails to adhere to standardized variable names, and does not appropriately specify units. The dataset header lacks key information, such as details about the analytical methods used and the complete list of contributors. It is true that they are included in the Zenodo version, but the file should stand by itself. Additionally, quality control flags for variables should be assigned to enhance the dataset's reliability and usability. Metadata completeness is crucial for long-term usability and re-usability. The file format should be open source (rather than an Excel file) and it would be much better distributed as a NetCDF file as it is interoperable with other platforms. The use of proprietary or non-open-source formats is a critical limitation, so a conversion to a format compatible with community standards is necessary. Since the authors intend to submit this dataset to SOCAT, which I consider an excellent decision as it is the current reference database for fCO2 measurements, it would be beneficial if the data product presented in ESSD could offer some added value or differentiation.

Specific comments (section and/or lines):

Reframe the title to reflect the true geographic scope, as the analysis is primarily focused on the Amazon River area rather than the entire east coast of South America.

Abstract 26-29: the order of the description should be homogeneous. For example, from river to ocean.

Abstract: North Brazil Current and Brazil Current are concepts not explained in the abstract. Not easy to follow.

Intro. 40. agree with *These regions present much higher temporal and spatial variability*. Therefore, this highly valuable snapshot only informs about spatial variability of a single season.

Intro 45. true sentence. The low number of observations in coastal waters is somewhat unexpected. Considering that coastal zones are, by definition, more accessible than the open ocean and offshore areas, it raises the question: why are these regions underrepresented in observational datasets? While addressing this issue is beyond the scope of this study, including a brief discussion or hypothesis to guide the reader would add valuable context.

Figure 1 is highly effective and well-designed, offering a clear overview of the study area. However, the inclusion of Pacific data skews the color bar and detracts from the manuscript's focus on equatorial South America in the Atlantic Ocean. It would be more appropriate to exclude the Pacific data and revise the figure to better align with the study's regional scope.

58. It represents one of the greatest environmental gradients on "the interface between" land and ocean in the world.

64 as the "Amazon" rainforest sequesters…

69. The references to the ANACONDAS (Mu et al., 2021) and Camadas Finas III (Araujo et al., 2017) campaigns lack both date signatures and spatial context. As a result, these terms may be largely unfamiliar to readers who are not specialized in regional studies, potentially hindering the manuscript's accessibility. Providing additional information about the campaigns' timing and geographical scope would help contextualize these references.

70. The manuscript mentions the link between these two systems, but it is unclear how this connection is made, as the schooner does not sail as far as Óbidos. If the link is indeed established, further clarification is needed, as it is not apparent from the current description. Could you please clarify whether the link is made, or revise the statement to reflect the actual coverage of the study?

71 I would delete "extensively"

75 in advance: The statement in paragraph 75 claiming that the Argo program could address the scarcity of fCO2 measurements in the medium term is not accurate. Currently, this is technically unfeasible, as the Argo program is designed

primarily for interior ocean monitoring. To measure fCO2, a surface-intensified approach would be required. The entire paragraph should be reassessed and revised to reflect the current limitations and the specific needs for fCO2 measurements.

90. As this dataset is based on underway measurements, you use kilometers to express the magnitude of the data, which is a valid option. However, it would be helpful to include additional information regarding the timing of the observations. For example, how many days of data were collected? Furthermore, how many different biomes were crossed during the survey? Providing this contextual information would enhance the understanding of the dataset's temporal and spatial coverage.

108-114. The port-to-port description could be better represented in a table format. This information, while useful, does not add significant value to the narrative and would be more concise and accessible in a tabular form.

In line 103, you mention 14,000 km, which is a considerable distance. However, the schooner sailed a total of 70,000 km. This discrepancy raises the question: why are the remaining 56,000 km not included in the analysis? While there may be valid reasons for excluding these data, offering an explanation would strength the manuscript's transparency.

Figure 2 does not accurately represent the full circuit, as it omits two branches. Please revise the figure to include these missing branches.

However, the dataset lacks integration with existing efforts (e.g., SOCAT) and does not demonstrate sufficient added value over what is already available. It would strengthen the manuscript to articulate why this dataset is unique and necessary in the context of global $fCO_2$ monitoring.

Section 2.2: You mention an important flow rate, which is often a bottleneck in underway systems on unconventional vessels. Could you please provide the model and specifications of the pump used? This information would be useful for understanding the system's limitations. Does the pump include a filter? Additionally, what is the maximum speed of the schooner at which the pump remains functional? Is the schooner's speed included in the dataset? If not, it should be, as this could be an important variable to consider.

Line 132: The term "accurately" could be removed.

Line 138: It is unclear to me why you do not have this data. In line 140, you mention a temperature difference, are you not talking about that in line 138?

Line 150: The sentences need to be reordered for clarity. It would be more effective to first describe the atmospheric air, then the reference gases, and finally the seawater. This would improve the logical flow of the section.

Line 160: When stating that the system is "cleaned regularly," it would be helpful to include the periodicity of the cleaning process. Additionally, does the intake circuit feature any physical barriers or filters to prevent the introduction of large particles? This should be clarified for completeness.

Line 185: You mention "yellow" in Figure 3, but no yellow is visible (at least to me) in the figure. Please revise the description to match the actual content of the figure.

Figure 3: The straight line in the "Raw" data does not convey meaningful information and appears to be an artifact caused by the connection of data points in the time series. This should be corrected to ensure the figure accurately represents the data.

Figure 4: Please include a scatter subplot that illustrates the 1:1 relationship between the discrete samples and the underway measurements. Use the timing of the bottle closure from the CTD surface samples, applying a time window of 2-3 minutes for the underway. Also, include the uncertainty associated with the discrete samples (5.7 µatm) and the range of underway measurements. The current format of this Figure 4, which combines all values into one plot, makes it difficult to discern patterns due to the large range of values. It would be clearer if the figure were split into three subplots: one for high values, one for the central values around 400, and another for low values.

Figure 5: While I appreciated the broad context provided by Figure 1, I would recommend a closer zoom for Figure 5 to focus specifically on the underway track. The current version leaves a significant amount of blank space that could be better utilized. The figure should be revised to eliminate this excess space and highlight the relevant data more clearly. Additionally, the arrangement of the subplots is unclear, and the "pyramid" layout may not be the most effective. I suggest reconsidering the layout for better clarity and visual coherence before publication.

3. Overview section: Ensure that the variables are always presented in the same order: temperature, salinity, and then fCO2.

Figure 7: Please add the locations of Obidos, Belem, and Macapa to the figure. Additionally, label some isobaths of bathymetry for reference.

Figures 8 & 9: It would be more efficient to include the area shown in Table 1 directly in the figures, thereby eliminating the need for the Table 1 itself. In my opinion, this will enhance the visual presentation and reduce redundancy.

Bathymetry: The bathymetry data is sourced from ETOPO2v2 and is colocalized along the ship track. However, including bathymetry values at 0 m or depths of 1-2 m raises some questions, as this would imply the ship was at or near the seabed (which, of course, we hope did not occur!). It may be more appropriate to group these shallow measurements or explore alternative methods for representing this data in a way that better reflects the actual conditions. Another comment about

bathymetry: I would strongly recommend adding the bathymetry data to the dataset. This would enable others to fully reproduce the analyses presented in the manuscript.

---

## Referee Comment (RC2)

ESSD Manuscript: "Exploring the $CO_2$ fugacity along the east coast of South America aboard the schooner Tara" - Oliver et al.

Time spent to review: 22h

Reviewer 2 comments:

The manuscript presents $CO_2$ fugacity data measured during the Tara expedition in the Tropical and Southwest Subtropical Atlantic Ocean, with a focus on the Amazonas River and the Brazilian coast. However, the manuscript contains several inconsistencies between figures and text, along with errors in names and terminology, which need to be corrected throughout. Key regional references are lacking, and the authors show limited understanding of the area. Additionally, the authors rely on outdated literature, some over 30 years old, which may no longer reflect the current local status and climatology.

Although the dataset covers the Caribbean to Argentina, the authors primarily focus the results and discussion on the Amazonas River plume and the North Brazil Current in the Equatorial Atlantic, with minimal discussion of the rest of the Southwest Atlantic Ocean. This narrow focus, combined with the lack of relevant references, leads to an over-interpretation of some parts of dataset and findings. Many of the findings and conclusions have already been published by other researchers in the same region, but this is not addressed clearly. Furthermore, there are several unsupported assumptions throughout the manuscript that need to be addressed.

Regarding data quality, the ESSD aims to ensure that high-quality and reproducible research data sets are included in permanent repositories. The authors mention issues with standard (STD) gases used for calibration but have not provided key data (e.g., pCO2air for STD calibration, in situ DIC and alkalinity data, or cruise metadata) for review. A major concern is the limited calibration with only two gases, one of which (502.3 ppm) is much lower than the values found in the river-ocean continuum (~2000 uatm), which falls outside the range, recommended in the SOCAT CookBook (2018). This raises doubts about the uncertainty of the data beyond the STD calibration range. Additionally, the authors report using only 13 discrete

samples to calibrate measurements over 14,000 km, which is statistically insufficient. This raises concerns, especially since only six in situ samples were taken during the longest part of the cruise. No statistical metrics are provided to support the calibration's accuracy and validation.

The authors claim there are no comparable datasets, focusing solely on the SOCAT database which excludes river measurements. I suggest they explore additional references to find relevant studies that might help:

- For the River-ocean continuum see: Valerio et al., 2018 (doi: https://doi.org/10.1364/OE.26.00A657) and the pre-print of Less et al., 2018 (https://doi.org/10.5194/bg-2018-465) - although the manuscript was withdrawn there is valuable information that authors can use.
- For Amazonas River Plume and North Brasil Current see: Monteiro et al., 2022 (doi: https://doi.org/10.1029/2022GB007385), Valério et al., 2021 (doi: https://doi.org/10.1016/j.csr.2021.104348), and others that authors should consult, for example, Ibánhez et al., 2016;  Lefèvre et al., 2014, 2017 and 2020.
- Other references about $CO_2$ fluxes, ocean $pCO_2$ and $fCO_2$ in the Brazilian coast can be found in the review paper Oliveira et al., 2022 (doi: http://dx.doi.org/10.21577/0100-4042.20170970).
- For the Vitória-Trindade Seamounts: Dynamics see: Napolitano et al., 2020 (doi: https://doi.org/10.1029/2020JC016731) and Silveira et al., 2020 (doi: https://doi.org/10.1007/978-3-030-53222-2_2). pCO2 data: request to Marinha do Brasil (Brazilian Navy).

In conclusion, while this manuscript has the potential to contribute to understanding $CO_2$ fluxes in the Tropical and Subtropical South Atlantic, the issues with data quality, limited analysis, and insufficient calibration prevent it from being ready for publication. I recommend a major revision, changing the manuscript focus specifically on the Amazonas River Plume and North Brazil Current. Due to the concerns raised, I do not recommend submitting the whole dataset to SOCAT v.2025, especially without addressing these issues.

Review by line:

Fig 1. Needs a geopolitical map, with countries borders, as in the manuscript refers to at least 4 different countries. Indicate where Martinique is. Correct the city name "Salvador" (not only in the figures but in all document). Increase figure axes font. I suggest to include different colours in the Tara path according with the Leg numbers presented in section 2.1.

54-55: "There are several reasons for this, including the reduced solubility of $CO_2$ at high temperatures, and the upwelling of deep waters rich in dissolved inorganic carbon (DIC) in the equatorial upwelling and along the coast." - Include reference.

58-59: "It represents one of the greatest environmental gradients on land and ocean in the world." - Include reference.

59-61: The Amazonas River plume is not constant in area and position along the year. Therefore, in this paragraph it would be good to have more information about seasonal variation of the Amazonas River discharge, which is intimately related with the rainy season and Intertropical Convergence Zone (ITCZ) position, as well as El Niño and La Niña years. This is important once you're analysing a period of transition between seasons. Literature suggestion: Kang et al., 2013 (doi: https://doi.org/10.1007/s13131-013-0269-5) and Lefèvre et al., 2014 (doi: https://doi.org/10.1002/2013JC009248).

62: Needs to describe the influence of salinity too as it was observed by other authors as one of the most important drivers to this area act as sink of $CO_2$. See recommended literature.

63-64: "Opposing this, the Amazon River outgasses nearly as much $CO_2$ as the rainforest sequesters on an annual basis." - Include reference. Do you have the numbers?

71-72: Although there is a need to better integrate the observations in the river and in the ocean, there are some studies that have been dedicated addressing these areas. Therefore, the references used here can be outdated. I suggest rephrasing or deleting this sentence. Please, also check the recommended references, and the project Carbon in the Amazon River Experiment (CAMREX).

52-74: I suggest a full review of this paragraph in view as key references are missing.

76-78: Are you referring to Argo or BCG-Argo floats? How much of the data gap in the surface ocean $fCO_2$ was covered by theBCG-Argo floats in the open ocean? Can you put this in numbers? Do you know why there are no BCG-Argo floats in these coastal areas?

78-81: I suggest you reorganize these sentences. There is a global decrease in the uploaded dataset of ship observations into SOCAT that is an interesting discussion to be included here. Start addressing why this might be happening and which region is more affected, then comment about the Brazilian coast. This raises an important point that just because the data is not in SOCAT it doesn't mean that there is no data in the region. This is particularly true for global south ocean.

87: Why is the equilibrator system more accurate than membrane systems?

91-94: The sentence is not completely true in the way it is written. There is the novelty of the sampling in the continuum using a sailboat. So, make sure to write this clearly, because it tends to give the idea that there is no data in these regions you refer to, which is not true. Also, it is very important to have in mind, not only for these lines but for all document, that this dataset is not filling the region data gaps, as it is just a snapshot of the conditions while the boat navigates in these particular areas. You can say that this data contributes to better understanding the area. Please, see the references suggested and provide new references that might be missing in there.

Introduction: You should focus the manuscript only in the Amazonas River Plume and NBC, as you give more attention to this area along the manuscript. Otherwise, you need to explore more and give more overview about the rest of the areas: Vitória-Trindade seamounts, Guanabara Bay, Lagoa dos Patos, and the rest of the Brazilian continental shelf.

95-96: Delete: "lesser studied".

104: In lines 90-91 you say that the measurements in the South America coast were between August and December 2021. And this line is saying until November 2021. Which one is correct?

107: 14.000 km is a long area, therefore only the dates don't give the real visualisation of the size of each Leg. It would be also very useful if you put the km or range of lat/long of each Leg. See suggestion for figure 1 about the colours.

110: Please, correct throughout the document (including figures). The correct name of the city is "Salvador".

113: Please, correct throughout the document (including figures). The correct name of the city is "Buenos Aires", it doesn't have hyphen.

117: Why did you choose this equilibrator system? Would be nice if you could include more details about the equilibrator system instead of just putting the reference, also with a figure/photo in figure 2.

118-119: "Currently, an equilibrator-based fCO2 system is the most reliable and accurate instrument to measure the in-situ fCO2 in seawater". Include a reference. Currently, the results of the $pCO_2$ Compare have not been released yet, is there other reference that has data to prove that?

119-120: "It is able to capture the fine scale variability of oceanic fCO2 by responding quickly to fCO2 changes in seawater" This seems very vague, please, include reference or rephrase.

123: How many square meters? Use precise numbers.

Figure 2: You could include a picture of the equilibrator system. Please, include where SBE 38 is in the scheme.

142: How the equilibrator air was dried? Please, include as much as information you have for this methodology as it can be replicated in future years. Including the sampling rates for STD gases, atmospheric and ocean.

144: I couldn't find the Annex.

149: Please, describe how these valves are controlled?

151: Why were these two standard gases chosen? There is a reference that suggests that or is something you are suggesting for the first time? As the mission would measure the river-ocean continuum, why didn't you choose a STD gas with a higher concentration to include the values between 0 and the maximum value found in the river, as recommended in SOCAT Cookbook (2018)?

154-155: "It is recommended to measure a complete set of standards every 3 hours." – Include reference. Can you explain more about these changes, in which leg it was made, for example.

163-164: How regularly the equilibrator was cleaned? There were other methods to avoid mud in the system?

168: Why did you proceed with this STD gas if it wasn't in the range reported by the supplier? Also, did you try to measure in a different analyser to see if the problem was the gas or your LICOR?

Figure 3: Please, increase the legend font. I suggest a scatter plot to make the comparison clearer and more realistic. Also, a statistical metric to support the relation between the datasets.

180: There is an inconsistency between the dates in the figure 3 (18-19/08/2021) and the main text (19-20/08/2021). My concerns about this correction is: i) it used just 7h of measurements, in just one point in the early stage of the campaign. Is this representative? Do you think it is possible to compare to another dataset or increase the calibration curve with the RPB measurements?

185-189: Which method? Compared the values and decreased one from the other? This is not a reliable calibration method for a long dataset as presented and with all unstable conditions for the gas cylinder. Please, provide a more reliable method that uses a significance range, calibration factor, or something that ensures your data is correctly calibrated and it is possible to replicate your calibration in other parts of the campaign.

193: Even though Metzl et al., 2024 provide the synthesis of SNAPO-CO2-v1 dataset, you need to be able to provide a brief explanation of the TA and DIC methodology used in your campaign, especially using these data to validate you $fCO_2$ dataset.

You also mentioned 17 samples for the surface, but you present only 13 in figure 4. Please, be clear in the text how many samples you used.

195: Provide which version of CO2SYS you used and the reference.

Figure 4: This figure gives a wrong perception of distance between the discrete samples. The figure could be sliced by leg to better visualize. In the current way it is not possible to address the values very easily. I also suggest a table where one column is the $fCO_2$ calculated by the CO2SYS and the other by the equilibrator.

205: "the continuous fCO2 compares very well to the one computed from the samples, especially after 26 September" – It is not possible to see this in the figure 4, and no statistical method was applied on the dataset to prove or support this sentence. Again, this is a long area with high variability, a comparison as presented in fig 4 is a weak assumption that your data has the required accuracy, especially for the river areas where you don't have neither discrete samples or the STD gas.

217: Please, check if your data matches the ones in the suggested references.

226: I strongly recommend you submit to SOCAT 2025 **only** the data for Amazonas River plume, as you didn't provide a reliable calibration and validation to the other parts of the dataset, especially in the rivers.

Figure 5: Please increase axes and colour bar font. I appreciate the consistency in the maps, however, it is not very easy to see the dataset variation in this. I suggest dividing by the legs. Please, include geopolitical map and include names (especially the ones you use in the text).

238: 36 is considered salty waters.

238: What does "recent ARP" mean in this context?

248: This can be due to rain in the land, which increases the river discharge, please check this and the references.

249: Change "maritime" to "marine". Please revise this in all document.

Figure 6: It gives more the idea of the places however the $fCO_2$ axes need to show with more numbers.

267: Please check the suggested references.

268: The correct name of the city is Rio de Janeiro.

270: Which is considered low salinities in Santos?

Figure 7: Please include the name of the cities and countries borders.

280: Please provide Lat/Lon of the center of the plume first here, and indicate in the figure 7.

282: "towards the Amazon" what? This seems incomplete.

Table 1 is not very informative as you could include this information in the text. If you follow the suggestion to focus the manuscript only in these areas, it would be interesting to see the difference of the mean $fCO_2$ for these regions which you could include in this table.

291: "follow well the relationship reported in Lefèvre et al., (2010)". Please provide a statistical metric that supports this, especially because the river region (brown) doesn't look like it fits well.

292: What does "NL" mean?

Discussion: Needs to be revised after including the key references and data validation metrics, and study area reduction to Amazonas River plume.

324: Tropical band is from 20°N to 20°S, where the tropics are. Perhaps you mean the equatorial area?

332: "changes in biogeochemical and biological properties." You didn't have biological data, so please include a reference for this sentence.

335: Provide the agreement with salinity.

343: "Nevertheless, while the large-scale variability of the fCO2 reflects the latitudinal temperature gradient" – include references that support your findings. See suggested references.

345-346: "Other river discharges reach the south Atlantic, such as the one of the Rio de la Plata." – Include references.

346-347: "These waters spread on the shelf and generate variability in salinity, suspended sediments and biological activity." – Include references.

352: Update reference.

354: Update reference.

361: Update reference.

361 – 362: "The source to sink transition is mainly driven by the switch from a respiration dominated system to a photosynthetic one." – Include references.

362-363: "Several factors impact the suspension of sediments in the water column and the development of phytoplankton, such as the bathymetry, winds and the tides." – include references. Include the rainy season, ITCZ position and, El Niño and La Niña occurrence impact.

365: Update reference.

388: "There are very little previously acquired data in the region that can be used for comparison." - Check references.

393: Delete "if they would exist." Check references first.

408: Delete "which had never been observed before." - Check references first.

408-409: Change "for the first time a sailboat equipped with a fCO2 system is sampling the river-ocean continuum". Otherwise this is an overinterpretation, especially because your river-ocean continuum data is not reliable.

416: "filling part of the data gap in the coastal regions of the South Atlantic Ocean", this is an over-interpretation, delete or rephrase.

420: Delete. The only mention of Guanabara bay is in the conclusion, and it is not easy to find the data of these regions in your results.

---

## Author Comment (AC1)

**Response to reviewer 1**

This manuscript presents an innovative methodology and a timely study, leveraging the capabilities of the *Tara* schooner to monitor CO$_2$ fugacity (fCO$_2$) in a key region for the global carbon cycle. The work demonstrates the potential of such expeditions for regional carbon dynamics studies and contributes valuable observations to the limited dataset from this area. Moreover, this represents an excellent initiative to expand the scope of the *Tara* schooner's activities—traditionally centered on marine biology and ecology—toward geoscience observations, as exemplified by the dataset presented in this manuscript. The manuscript is well-written, well-structured, and includes high-quality figures. The article is worthy of publication and appropriate to support the publication of a data set, but the dataset itself exhibits significant shortcomings that must be first addressed. Important modifications are required to ensure the dataset meets the standards of openness, metadata completeness, and long-term usability. I recommend a major revision, primarily to address issues with the data product, as outlined below. Additionally, there are inconsistencies in the framing and presentation that detract from the overall clarity and impact of the work, but they should be easily ameliorated.

*First, we would like to thank the reviewer for his positive comments on the paper, and for this detailed, complete and constructive review. We agree that the dataset presentation and associated metadata needed a little bit more structuring and work that now has been done. We believe that this review made the new version of the manuscript significantly better, so we thank the reviewer for taking the time to read the paper in depth and comment it. We will address each comment, and indicate in orange the changes made in the manuscript.*

The current form of the data product significantly weakens the manuscript's suitability for publication in ESSD, a journal known for its rigorous standards in data quality and accessibility. The dataset lacks metadata, the data file itself it has an untraceable naming ("*CO2Tara.xlsx*" -sic-), fails to adhere to standardized variable names, and does not appropriately specify units. The dataset header lacks key information, such as details about the analytical methods used and the complete list of contributors. It is true that they are included in the Zenodo version, but the file should stand by itself. Additionally, quality control flags for variables should be assigned to enhance the dataset's reliability and usability. Metadata completeness is crucial for long-term usability and re-usability. The file format should be open source (rather than an Excel file) and it would be much better distributed as a NetCDF file as it is interoperable with other platforms. The use of proprietary or non-open-source formats is a critical limitation, so a conversion to a format compatible with community standards is necessary. Since the authors intend to submit this dataset to SOCAT, which I consider an excellent decision as it is the current reference database for fCO2 measurements,

it would be beneficial if the data product presented in ESSD could offer some added value or differentiation.

*We completely agree with this comment, and we apologize for not having done it before the first submission. It is entirely true that the file should stand by itself, and should be in an open-source format. Here, we propose to include in Zenodo a full additional metadata file, describing the system and the uncertainties of each sensor, as well as the dates of calibration, as is necessary for the submission to the SOCAT database. In addition, we included a header in the dataset, as suggested, indicating the purpose of the dataset, the main contributors, the method of acquisition and the region/time-period sampled. We will change the format of the dataset to csv and netcdf to provide more variety in the formats (both open-source).*
*Regarding the differentiation, this is a good comment. We propose here to add the bottom depth and the measured wind speed, as both parameters are very useful for the interpretation of the dataset. The wind speed is essential for calculating the $CO_2$ flux, and the bottom depth is closely related to the fCO2 variability as analyzed in the document. The data have been submitted to SOCAT, the added value here is also the qualification and validation of the dataset as well as comparison to other in-situ data, well necessary in coastal environments.*

**Specific comments (section and/or lines):**

Reframe the title to reflect the true geographic scope, as the analysis is primarily focused on the Amazon River area rather than the entire east coast of South America.
*Although the primary focus of the paper is the equatorial area close to the Amazon plume and the estuary, the dataset and its description cover a much wider area, which is roughly 2/3 of the meridional length of South America, from the Little Antilles to the border of Uruguay. If the reviewer agrees, we would therefore prefer to keep the title as is to avoid making it too long. Otherwise, we would suggest: Exploring the $CO_2$ fugacity along the east coast of South America aboard the schooner Tara: the Amazon River and beyond*

Abstract 26-29: the order of the description should be homogeneous. For example, from river to ocean.
*Thank you for the good suggestion, we modified the abstract, starting from the river, then the river plume and finally the open ocean, moving south. Observations revealed a wide range of fCO2 values, from up to up to 3000 $\mu$atm in the river to a minimum of 42 $\mu$atm downstream of the plume, where values were notably lower than atmospheric levels. South of the estuary, the $fCO_2$ of the North Brazil Current's waters (0-9°S) exceeds 400 $\mu$atm while along the Brazil Current (10-30°S), $fCO_2$ is around 400 $\mu$atm and decreases with temperature and distance from the equator.*

Abstract: North Brazil Current and Brazil Current are concepts not explained in the abstract. Not easy to follow.

*We believe that these currents are accepted concepts, but we added the bands of latitudes concerned for clarity.*

Intro. 40. agree with *These regions present much higher temporal and spatial variability*. Therefore, this highly valuable snapshot only informs about spatial variability of a single season. *Thank you, we fully agree.*

Intro 45. true sentence. The low number of observations in coastal waters is somewhat unexpected. Considering that coastal zones are, by definition, more accessible than the open ocean and offshore areas, it raises the question: why are these regions underrepresented in observational datasets? While addressing this issue is beyond the scope of this study, including a brief discussion or hypothesis to guide the reader would add valuable context.
*We agree, and we find the low number of observations quite concerning. One explanation is that some of these coastal zones are not that easy to sample, either due to distance to major ports, or because of accessing permits to perform the measurements (since most of them are conducted in EEZ). We added this hypothesis in the conclusion.* The limited number of observations could be due to the complicated access to some of these regions (distance from major port) and to the difficulty of obtaining sampling permits.

Figure 1 is highly effective and well-designed, offering a clear overview of the study area. However, the inclusion of Pacific data skews the color bar and detracts from the manuscript's focus on equatorial South America in the Atlantic Ocean. It would be more appropriate to exclude the Pacific data and revise the figure to better align with the study's regional scope.
*Agreed, here is the new version of Figure 1 without the Pacific data. The colorbar was kept the same because it is centered on 400 µatm that is approximately the atmospheric value, so that the reader can easily identify visually potential sink and sources.*

[Figure]

It represents one of the greatest environmental gradients on "the interface between" land and ocean in the world.
*OK, added.*

64 as the "Amazon" rainforest sequesters...
*OK, added.*

The references to the ANACONDAS (Mu et al., 2021) and Camadas Finas III (Araujo et al., 2017) campaigns lack both date signatures and spatial context. As a result, these terms may be largely unfamiliar to readers who are not specialized in regional studies, potentially hindering the manuscript's accessibility. Providing additional information about the campaigns' timing and geographical scope would help contextualize these references.

*Thank you for your perspective. We added the years of the cruise and their focus. On the other hand, oceanographic studies, carried out in particular during the ANACONDAS (in 2011, 2012 and 2013, Mu et al., 2021) and Camadas Finas III (October 2012, Araujo et al., 2017) campaigns that focused on the ARP development, maximum extension and early decay have shown the extent of $CO_2$ undersaturation in the ARP.*

The manuscript mentions the link between these two systems, but it is unclear how this connection is made, as the schooner does not sail as far as Óbidos. If the link is indeed established, further clarification is needed, as it is not apparent from the current description. Could you please clarify whether the link is made, or revise the statement to reflect the actual coverage of the study?

*It is very true that Tara did not sail as far as Obidos, but slightly inland of Macapa. However, the connection is made, not only because Macapa is fully inland with no salt water influence, but also because the previous and key study of Sawakuchi et al stopped at Macapa (they extended the river from Obidos to Macapa). Were are therefore adding the missing link, Macapa-Open ocean.* To improve clarity, we added in the manuscript that the missing link is the amazon estuary, sampled by the schooner. *However, the estuary, which is the link between these two systems is little known, if at all as riverine observations stopped in Macapá.*

71 I would delete "extensively". *OK, done*.

75 in advance: The statement in paragraph 75 claiming that the Argo program could address the scarcity of fCO2 measurements in the medium term is not accurate. Currently, this is technically unfeasible, as the Argo program is designed primarily for interior ocean monitoring. To measure fCO2, a surface-intensified approach would be required. The entire paragraph should be reassessed and revised to reflect the current limitations and the specific needs for fCO2 measurements.

*Thank you for this comment, this is a very interesting and debated issue in the community. The mention of the Argo program was certainly not done to signify that a surface-intensified approach was not deemed necessary, but to mention added capability to observe the near surface ocean (albeit indirectly in the case of BGC Argo float) that is being implemented. In data-scarce area, such as the Southern Ocean, using Argo float-derived $CO_2$ fluxes brought a new understanding of the area and showed contributed to identify limitations in our observations. We completely agree that a surface-intensified approached should be strongly encouraged when measuring $fCO_2$, and Argo floats enriches this information information by bringing complementary biogeochemical*

*parameters that are worth looking at. We therefore rephrased our paragraph. While measurements of biogeochemical parameters in the open ocean have increased in recent years owing to the development of the biogeochemical Argo program, it is not the case for biogeochemical measurement on the shelves and continental margins. Moreover, continuous surface fugacity of $CO_2$ ($fCO_2$) measurements carried out on ships remain the most accurate way to asses $CO_2$ fluxes and are still too sparse.*

As this dataset is based on underway measurements, you use kilometers to express the magnitude of the data, which is a valid option. However, it would be helpful to include additional information regarding the timing of the observations. For example, how many days of data were collected? Furthermore, how many different biomes were crossed during the survey? Providing this contextual information would enhance the understanding of the dataset's temporal and spatial coverage.
*We agree, thank you for the suggestion, we included the number of days (81 days).*

108-114. The port-to-port description could be better represented in a table format. This information, while useful, does not add significant value to the narrative and would be more concise and accessible in a tabular form.
*Agreed, thank you very much for the good suggestion. This section has been modified to include the table and is now: The dataset presented in this study focuses on the underway data collected during the legs 5, 6, 7, 8 and 9 of the Mission Microbiome (Table1). The dataset covers 14,000 km and stops on 25 November, as the authorization to sample in the exclusive economic zone of Uruguay was not obtained. Attempts were subsequently made to restart the system during leg 11, but these were aborted, as the conditions of the standard gas cylinders did not allow the same accuracy to be achieved.*

In line 103, you mention 14,000 km, which is a considerable distance. However, the schooner sailed a total of 70,000 km. This discrepancy raises the question: why are the remaining 56,000 km not included in the analysis? While there may be valid reasons for excluding these data, offering an explanation would strength the manuscript's transparency.
*The remaining 56,000 km are not included because the pCO2 system stopped working. While crossing the Drake Passage (stage 11), one of the gas cylinders used for calibration leaked and emptied, preventing us from calibrating the data correctly. Due to COVID and strict regulations, it was almost impossible to ship another gas cylinder in time to continue the measurements. We added a short version of this explanation to the manuscript, see modification in the previous comment.*

Figure 2 does not accurately represent the full circuit, as it omits two branches. Please revise the figure to include these missing branches. *We are very sorry, but we don't see which branches are missing. It is true that the branches for the 0 and 502.3 ppm standards are merged into one under the name 'standards' to avoid overloading the schematic. Then there is the atmospheric air branch and the branch coming from the equilibrator, all connected to a valve that control the air sent to the LICOR analyzer. Thus, this adds up to*

*four branches. We added the TSG exit branch, and made clearer that there are two standards.*

[Figure]

However, the dataset lacks integration with existing efforts (e.g., SOCAT) and does not demonstrate sufficient added value over what is already available. It would strengthen the manuscript to articulate why this dataset is unique and necessary in the context of global $fCO_2$ monitoring.

*We agree, thank you for pointing it out. We tried first to point out what this dataset brings out on top of the dataset submitted to SOCAT (ancillary data very useful for the interpretation and quality control of the data), and also to show how this dataset is unique (the first equilibrator-based system on a sailboat, unique region, diverse gradients) and necessary (overall lack of data etc...). In the manuscript (section 2.5), we included:* In the dataset, ancillary data are added (wind speed at 10 m, bottom depth) to offer a more detailed interpretation of the data. Wind speed was measured by a Gill anemometer at the top of the mast (27 m), and then adjusted to 10 m using a logarithmic relationship (Tennekes, 1973). This dataset addresses the overall lack of data identified by SOCAT, by covering diverse environmental gradients with a high-resolution sampling. The use of the schooner highlights the potential of non-traditional platforms for collecting high-quality data in challenging environments, complementing traditional research vessels.

Section 2.2: You mention an important flow rate, which is often a bottleneck in underway systems on unconventional vessels. Could you please provide the model and specifications of the pump used? This information would be useful for understanding the system's limitations. Does the pump include a filter? Additionally, what is the maximum speed of the schooner at which the pump remains functional? Is the schooner's speed included in the dataset? If not, it should be, as this could be an important variable to consider. *We agree: on Tara to avoid the bottleneck there are two pumps feeding the underway system. The pump is a Shurflow probait master 4 from Penatair. The maximum flow rate is 12L/min. After the pump the circuit includes a*

*debubbler and a large particles filter, we added this information to the manuscript: It then goes through a large particles filter and enters a debubbler to remove most of the bubbles that can be caused by such shallow water intake, especially in rough seas. Tara is not a very fast sailing sailboat, so we never reached a speed at which the pump couldn't work anymore. For example, Tara's average speed is 6kn, while a research vessel's speed is 10kn. One of the issues with sailing vessels and the shallow intake is excess bubbles entering the system when the sea is rough. Fortunately, these conditions were not encountered here.*

Line 132: The term "accurately" could be removed. *Ok, removed.*

Line 138: It is unclear to me why you do not have this data. In line 140, you mention a temperature difference, are you not talking about that in line 138? *Unfortunately, there was no temperature sensor in the equilibrator. This is why we used the TSG temperature, which, based on how the system was installed, should provide a rather good estimate of the equilibrator's temperature. In line 140, we talk about the temperature difference between the hull temperature sensor (SBE38, true SST) and the TSG temperature (SBE45). On a sailboat, the difference is very small, but on a research vessel the difference can be much larger than 0.1°C, and in this case the $fCO_2$ data needs to be discarded (according to the sampling best practices). We tried to make this clearer by modifying Figure 2, and including the SBE38 in the schematic so that it is possible to identify visually the two temperature sensors. In the manuscript: the temperature difference between the hull sensor (SBE38) and the TSG is small (always below 0.1 °C and averaging 0.07°C, Figure 2).*

Line 150: The sentences need to be reordered for clarity. It would be more effective to first describe the atmospheric air, then the reference gases, and finally the seawater. This would improve the logical flow of the section. *We agree, thank you, we modified: Through a system of valves, four circuits are operated, one for the atmospheric air, one for each of the two reference gases, and one for the air equilibrated with seawater.*

Line 160: When stating that the system is "cleaned regularly," it would be helpful to include the periodicity of the cleaning process. Additionally, does the intake circuit feature any physical barriers or filters to prevent the introduction of large particles? This should be clarified for completeness. *This is also true. the system was cleaned at each stopover (5 times), and each time the boat exited a major river (two additional times leaving Macapá and Belém). There is a large particle filter on the line, this has been added to the description of the system following the comment on the pump. We propose to add in the manuscript: The equilibrator was cleaned at each stopover, and each time the ship exited a major river (so 7 times in total) to avoid the buildup of mud, and the system therefore recorded data during the whole time spent in the Amazon River.*

Line 185: You mention "yellow" in Figure 3, but no yellow is visible (at least to me) in the figure. Please revise the description to match the actual content of the figure.
*We agree that it is not easy to describe this color, we changed yellow by light brown in both the text and the legend of the figure, hopefully it improved clarity.*

Figure 3: The straight line in the "Raw" data does not convey meaningful information and appears to be an artifact caused by the connection of data points in the time series. This should be corrected to ensure the figure accurately represents the data. *We agree and removed the connection between the points.*

[Figure]

Figure 4: Please include a scatter subplot that illustrates the 1:1 relationship between the discrete samples and the underway measurements. Use the timing of the bottle closure from the CTD surface samples, applying a time window of 2-3 minutes for the underway. Also, include the uncertainty associated with the discrete samples (5.7 µatm) and the range of underway measurements. The current format of this Figure 4, which combines all values into one plot, makes it difficult to discern patterns due to the large range of values. It would be clearer if the figure were split into three subplots: one for high values, one for the central values around 400, and another for low values. *Following your comment, we revised Figure 4. The figure now has 3 panels as suggested, but we kept the first two as previously, because they show the full range of values, both in the upper values of the Amazon River (top panel) and the lower values of the plume (2nd panel, now in the middle). To address the comment, the third panel focuses on the 350-450 $fCO_2$ range, which includes most of the underway and discrete measurements. On this panel, we also added the discrete samples uncertainty of 5.7 µatm as error bars. We included the scatter plot as a fourth panel. For the scatter plot, some interpretation needs to be considered. For stations 36abc, the system was calibrating while the sample was taken. In this variable region, we do not feel comfortable either extrapolating or using a value measured 15min before or later. Last piece of information, we analyzed the difference between the salinity of the bottle from a salinity sample and the salinity measured by the Rosette's CTD, the difference can be up to 3 pss from station 35 to 42, due to the high surface variability of the region. For all these reasons, we prefer to focus on stations 45 onwards to assess the accuracy. We integrated these explanations in a more concise form in the manuscript (legend of Figure 4), but we believed the reviewer might be interested in more details.*

[Figure]

*Figure 4 : Time-series of surface $fCO_2$ from 18/08/2021 to 25/11/2021, for the full range of values (a) for only oceanic values (b) for values between 350 and 450 µatm (c). The dots indicate the $fCO_2$ inferred from the DIC/TA water samples for stations 35 to 49, with error bars of 5.7 µatm to represent the uncertainty of the chemical formulas. Scatter plot of the underway $fCO_2$ and the $fCO_2$ inferred from the DIC/TA samples, for $fCO_2$ values ranging between 350 and 450 µatm (d). The green dot indicates a salinity difference between the CTD sensor and the sample from bottle of more than 0.5. The $fCO_2$ system was measuring the standards gases for calibration during stations 36abc and 39, these stations are therefore not represented in (d).*

Figure 5: While I appreciated the broad context provided by Figure 1, I would recommend a closer zoom for Figure 5 to focus specifically on the underway track. The current version leaves a significant amount of blank space that could be better utilized. The figure should be revised to eliminate this excess space and highlight the relevant data more clearly. Additionally, the arrangement of the subplots is unclear, and the "pyramid" layout may not be the most effective. I suggest reconsidering the layout for better clarity and visual coherence before publication. *Thank you, we completely agree, here is a revised version of Figure 5.*

[Figure]

Overview section: Ensure that the variables are always presented in the same order: temperature, salinity, and then fCO2. *Thank you, we modified this section and try to*

*follow the order you suggested. However, when the fronts were driven by salinity, we preferred to start by salinity, then temperature and finally fCO2.*

Figure 7: Please add the locations of Obidos, Belem, and Macapa to the figure. Additionally, label some isobaths of bathymetry for reference. *Thank you, done. We didn't add Obidos because it is not in the area represented on the map.*

[Figure]

Figures 8 & 9: It would be more efficient to include the area shown in Table 1 directly in the figures, thereby eliminating the need for the Table 1 itself. In my opinion, this will enhance the visual presentation and reduce redundancy.
*Thank you for this good suggestion. We removed the table and added another panel to figure 9 (Figure 8 already had 4 panels) directly showing the areas on a map. Here is the revised Figure 9:*

[Figure]

Bathymetry: The bathymetry data is sourced from ETOPO2v2 and is colocalized along the ship track. However, including bathymetry values at 0 m or depths of 1-2 m raises some questions, as this would imply the ship was at or near the seabed (which, of course, we hope did not occur!). It may be more appropriate to group these shallow measurements or explore alternative methods for representing this data in a way that better reflects the actual conditions. Another comment about bathymetry: I would strongly recommend adding the bathymetry data to the dataset. This would enable others to fully reproduce the analyses presented in the manuscript.
*Thank you again for an interesting comment. We agree that the bathymetry ETOPO2v2 does not resolve well the river, and for further analysis of the river part of the dataset a dedicated*

*bathymetry should be used, such as the one described in Fassoni-Andrade et al., Comprehensive bathymetry and intertidal topography of the Amazon estuary, (2021). ESSD. https://doi.org/10.5194/essd-13-2275-2021. Nonetheless, this is a bit out of scope for this study and the grouping in Figure 9 is still relevant at 0-order, even though the uncertainties are large. We modified the manuscript to show this limitation of the dataset, and also refer to a more accurate one if the user is interested. In the discussion:* *Their "River Zone", for depth below 5 m, indeed corresponds to a salinity of 0. Nevertheless, we observe significant variability of fCO2 even if the salinity does not change anymore. This region was not investigated by these studies that focused further offshore of the mouth. For the region, the bathymetry used here is not adapted anymore, and a specific Amazon estuary bathymetry should be used for further studies (such as Fassoni-Andrade et al., 2021).*

*We agree with the suggestion of adding the bathymetry to the dataset. Both the bathymetry and the measured winds are important relevant variable, that are not included in the SOCAT database and that we will include in this dataset.*

---

## Author Comment (AC2)

**Response to reviewer 2**

*Dear Reviewer,*
*Thank you for taking the time to carefully review our manuscript. We appreciate your detailed comments, which provide valuable insight to improve the quality of our work. We will address each comment, and indicate in* orange *the changes made in the manuscript.*

The manuscript presents CO$_2$ fugacity data measured during the Tara expedition in the Tropical and Southwest Subtropical Atlantic Ocean, with a focus on the Amazonas River and the Brazilian coast. However, the manuscript contains several inconsistencies between figures and text, along with errors in names and terminology, which need to be corrected throughout. Key regional references are lacking, and the authors show limited understanding of the area. Additionally, the authors rely on outdated literature, some over 30 years old, which may no longer reflect the current local status and climatology.

Although the dataset covers the Caribbean to Argentina, the authors primarily focus the results and discussion on the Amazonas River plume and the North Brazil Current in the Equatorial Atlantic, with minimal discussion of the rest of the Southwest Atlantic Ocean. This narrow focus, combined with the lack of relevant references, leads to an over-interpretation of some parts of dataset and findings. Many of the findings and conclusions have already been published by other researchers in the same region, but this is not addressed clearly. Furthermore, there are several unsupported assumptions throughout the manuscript that need to be addressed.

*We acknowledge the importance of ensuring consistency between figures, text, and terminology. We have thoroughly reviewed the manuscript and corrected any discrepancies or errors in naming conventions and terminology.*
*We appreciate your suggestion to include more regional references. While we agree that incorporating relevant and recent literature is essential, we have carefully evaluated the references provided in your comments and included those that enhance the context of our work. However, some of the suggested references do not align directly with the focus of this study. We aim to ensure that the manuscript reflects a balanced and pertinent citation of the literature.*
*The data collected in the Southwest Atlantic Ocean are indeed less novel, and we provided less scientific background for them. However, they are an integral part of the same dataset, validated using the same methodology, and therefore fully belong in this manuscript.*
*We agree that the most significant contribution of our dataset is to the study of the Amazon River plume and the land-sea continuum. While we acknowledge that we could have included a broader range of references on this topic, many of the suggested publications provide limited insights into actual pCO$_2$ variability in the Amazon River plume. Additionally, there remains a scarcity of published, accessible pCO$_2$ data in this region,*

*particularly during the late flood season, with a stronger focus on the North Brazil Current and the Pará River plume.*

Regarding data quality, the ESSD aims to ensure that high-quality and reproducible research data sets are included in permanent repositories. The authors mention issues with standard (STD) gases used for calibration but have not provided key data (e.g., pCO2air for STD calibration, in situ DIC and alkalinity data, or cruise metadata) for review. A major concern is the limited calibration with only two gases, one of which (502.3 ppm) is much lower than the values found in the river-ocean continuum (~2000 uatm), which falls outside the range, recommended in the SOCAT CookBook (2018). This raises doubts about the uncertainty of the data beyond the STD calibration range. Additionally, the authors report using only 13 discrete samples to calibrate measurements over 14,000 km, which is statistically insufficient. This raises concerns, especially since only six in situ samples were taken during the longest part of the cruise. No statistical metrics are provided to support the calibration's accuracy and validation.

*We acknowledge that not all data can meet the highest quality standards, even within SOCAT guidelines, which explicitly accept data of varying quality levels. In regions with large variability, such as the Amazon River–ocean continuum and plume, a slightly reduced precision does not significantly impact the dataset's scientific value.*
*The choice of reference gases was constrained by availability but was also made to optimize accuracy for the most common oceanic surface water conditions in the study area. While we recognize that this calibration approach introduces higher uncertainty at the extreme $pCO_2$ values observed in the river–ocean continuum, it remains within an acceptable range for scientific interpretation. Unfortunately, we currently lack the means to quantitatively assess the uncertainty specifically at these high $pCO_2$ levels. However, we have reported uncertainty estimates for a more typical oceanic range, that is the majority of the dataset. Regarding independent validation with DIC and TA measurements, we acknowledge the limited number of usable samples. The discrepancy between the 13 retained and the 17 collected is due to unsatisfactory sample quality in some cases (e.g., riverine waters) and the absence of simultaneous $pCO_2$ measurements. While we recognize that 13 samples are fewer than ideal, logistical constraints and the practical challenges of recommended sampling protocols made broader coverage difficult across the Southwest Atlantic expedition legs. Nonetheless, these values, combined with the $pCO_2$ air measurements, provide an additional layer of validation to support our accuracy assessment. It is an added value, as SOCAT for example does not require comparison with DIC/TA samples.*

The authors claim there are no comparable datasets, focusing solely on the SOCAT database which excludes river measurements. I suggest they explore additional references to find relevant studies that might help:
•       For the River-ocean continuum see: Valerio et al., 2018 (doi: https://doi.org/10.1364/OE.26.00A657) and the pre-print of Less et al., 2018

(https://doi.org/10.5194/bg-2018-465) - although the manuscript was withdrawn there is valuable information that authors can use.
•       For Amazonas River Plume and North Brasil Current see: Monteiro et al., 2022 (doi:   https://doi.org/10.1029/2022GB007385),   Valério   et   al.,   2021   (doi: https://doi.org/10.1016/j.csr.2021.104348), and others that authors should consult, for example, Ibánhez et al., 2016; Lefèvre et al., 2014, 2017 and 2020.
•       Other references about CO2 fluxes, ocean pCO2 and fCO2 in the Brazilian coast can   be   found   in   the   review   paper   Oliveira   et   al.,   2022   (doi: http://dx.doi.org/10.21577/0100-4042.20170970).
•       For the Vitória-Trindade Seamounts: Dynamics see: Napolitano et al., 2020 (doi: https://doi.org/10.1029/2020JC016731)   and   Silveira   et   al.,   2020   (doi: https://doi.org/10.1007/978-3-030-53222-2_2). pCO2 data: request to Marinha do Brasil (Brazilian Navy).

*We appreciate the reviewer's suggestion to explore additional references; however, we respectfully clarify that our manuscript does not focus on riverine $pCO_2$ measurements alone. Instead, it emphasizes the river–ocean continuum and the Amazon River plume, with only limited data collected within the river itself. Therefore, while some of the suggested references provide valuable insights into riverine processes, they do not directly align with the scope of our study.*

*Regarding the suggested references:*

- *River-focused studies (e.g., Valerio et al., 2018; Less et al., 2018) primarily investigate seasonal variability in river properties. We appreciate the reviewer's suggestion and have cited Valerio et al. (2018) in our manuscript. Their discrete measurements, taken in April 2017, provide valuable context, though a direct comparison with our data from September 2021 is not possible. While these studies offer important insights by extending to the river mouth, they do not explore the connection with the Amazon River Plume beyond the estuary.*
- *Amazon plume studies: Valerio et al. (2021) utilizes $pCO_2$ data from three Anacondas cruises to develop a model based on an early version of SMOS sea surface salinity (SSS). However, it is not a data paper (unlike Mu et al. 2021, which we already reference). Additionally, the Anacondas cruises do not overlap with our dataset temporally or spatially—the closest observations to the estuary were from July 2012, outside our study period, and most data were collected much farther downstream off French Guiana in September–October. Thus, this study does not provide a direct comparison for our results.*
- *Monteiro et al. (2022) presents an interesting analysis based on the SOCAT database, which we reference in our manuscript. However, this dataset includes very few measurements in the area and season relevant to our study, limiting its applicability to our findings.*
- *Other suggested references: Some of the suggested works (e.g., Ibanhez et al. 2016, Lefèvre et al. 2017) are already cited in our manuscript. Lefèvre et al. (2014) provides*

*data at 38°W off French Guiana with one August crossing, which, while valuable, does not significantly enhance the contextual relevance of our dataset due to limited overlap. Lefèvre et al. (2020) focuses on the NECC at 8°N, an entirely different region.*
*For the Southwest Atlantic, the references provided are of limited relevance to our study:*

- *Oliveira et al. (2022) is written in Portuguese, provides a general overview of Brazilian shelf datasets but lacks clear data coverage details beyond publicly available SOCAT sources.*
- *Napolitano et al. (2020) and Silveira et al. (2020) focus on ocean dynamics near the Vitória-Trindade Seamounts, which, while scientifically interesting, are not central to our dataset. While these processes were one motivation for Tara's sampling in the region, there is no direct evidence that these phenomena influenced the $pCO_2$ measurements collected. We therefore included them in the introduction to provide context.*
- *Silveira et al. (book chapter) presents a summary that includes model reanalysis but does not contribute substantially to the analysis of in situ $pCO_2$ observations.*

*Regarding the data from Marinha do Brasil, we acknowledge that these datasets could be valuable. However, since they are not publicly available (requests for access have not been granted), we are unable to incorporate them into our analysis.*

*In summary, while we appreciate the reviewer's suggestions, many of these references do not directly address the $pCO_2$ variability in the Amazon River–plume continuum or the specific oceanic regions and seasons covered by our dataset. Nevertheless, we will carefully reconsider our reference list and ensure that we include any additional relevant studies where appropriate.*

*We now provide a response to the detailed comments.*

Review by line:

Fig 1. Needs a geopolitical map, with countries borders, as in the manuscript refers to at least 4 different countries. Indicate where Martinique is. Correct the city name "Salvador" (not only in the figures but in all document). Increase figure axes font. I suggest to include different colours in the Tara path according with the Leg numbers presented in section 2.1.

*Thank you for the helpful suggestions. We have corrected the city name to Salvador throughout the manuscript. In figure one, we indicated where Martinique is, and corrected "Salvador da Bahia" in Salvador. Regarding the addition of country borders we followed the explicit recommendation of the journal: "In order to depoliticize scientific articles, authors should avoid the drawing of borders" (ESSD submission guidelines, section maps & aerials: https://www.earth-system-science-data.net/submission.html#figurestables ). Thank you for your suggestion regarding the use of different colors for the Legs. However, since our analysis does not focus on interpreting the data leg by leg, we believe that adding this information to the figure would unnecessarily clutter it without serving a specific purpose in the manuscript.*

[Figure]

*Figure 1: Revised version of Figure 1*

54-55: "There are several reasons for this, including the reduced solubility of CO2 at high temperatures, and the upwelling of deep waters rich in dissolved inorganic carbon (DIC) in the equatorial upwelling and along the coast." - Include reference.

*Thank you, we included a reference to a paper showing and studying the $CO_2$ outgassing in tropical waters, and the role of the equatorial upwelling: Andrié, C., Oudot, C., Genthon, C., and Merlivat, L.: CO2 fluxes in the tropical Atlantic during FOCAL cruises, Journal of Geophysical Research: Oceans, 91, 11741–11755, https://doi.org/10.1029/JC091iC10p11741, 1986.*

58-59: "It represents one of the greatest environmental gradients on land and ocean in the world." - Include reference.

*Thank you, we added a reference to Araujo et al., (2017).*

59-61: The Amazonas River plume is not constant in area and position along the year. Therefore, in this paragraph it would be good to have more information about seasonal variation of the Amazonas River discharge, which is intimately related with the rainy season and Intertropical Convergence Zone (ITCZ) position, as well as El Niño and La Niña years. This is important once you're analysing a period of transition between seasons. Literature suggestion: Kang et al., 2013 (doi: https://doi.org/10.1007/s13131-013-0269-5) and Lefèvre et al., 2014 (doi: https://doi.org/10.1002/2013JC009248).

*Thank you for this suggestion. We fully acknowledge the seasonal and interannual variability of the Amazon River plume, however, as this manuscript is primarily a data paper rather than a process-focused analysis, we have kept the discussion concise, referring readers to Olivier et al. (2024) and references therein for a more detailed analysis of the 2021 conditions and broader climatological context. That said, we now explicitly state that the cruise took place during a period of decreasing Amazon outflow, following one of the largest Amazon flood events on record, a bit later in the manuscript's introduction: This novel dataset presents 14,000 km of fCO2 measurements over 98 days between August to end of November 2021, primarily along the South American coast, and marking the first repeated sampling of the AROC. The cruise took place in a period of decreasing rive outflow, following one of the largest Amazon flood events on record. Freshwater transport was strongly directed toward the Caribbean, with comparatively less Amazon-derived freshwater reaching the NECC and central Atlantic (Olivier et al., 2024b).*
 *We appreciate the references provided and reviewed them to ensure that relevant aspects of seasonal variability are appropriately acknowledged.*

62: Needs to describe the influence of salinity too as it was observed by other authors as one of the most important drivers to this area act as sink of CO2. See recommended literature.

*Thank you, we added this influence of salinity: combined with low salinities (Ibánhez et al., 2016; Lefévre et al., 2010)*

63-64: "Opposing this, the Amazon River outgasses nearly as much CO2 as the rainforest sequesters on an annual basis." - Include reference. Do you have the numbers?

*We included a reference to Sawakuchi et al., 2017 and Richey et al., 2002. We also recommend, seeing your interest in the question, this communication from the journal frontiers:*
*https://www.frontiersin.org/news/2017/05/15/frontiers-in-marine-science-study-finds-amazon-river-carbon-dioxide-emissions-nearly-balance-terrestrial-uptake*

71-72: Although there is a need to better integrate the observations in the river and in the ocean, there are some studies that have been dedicated addressing these areas. Therefore, the references used here can be outdated. I suggest rephrasing or deleting this sentence. Please, also check the recommended references, and the project Carbon in the Amazon River Experiment (CAMREX).

*Thank you for bringing the CAMREX references to our attention, we find the project very interesting. While these studies are indeed valuable, they primarily focus on riverine waters and do not extend to the transition zone between the river and the ocean. As we highlight in our manuscript, there is still a lack of recent surveys that fully capture the river-ocean continuum, particularly during this season, and this is the main focus of this paragraph. In particular, few studies have documented the transition from high $pCO_2$ river waters to the low surface values observed in the Amazon River Plume. We have reviewed the suggested references and will incorporate any relevant insights to ensure our discussion is as up-to-date as possible.*

52-74: I suggest a full review of this paragraph in view as key references are missing.

*We reviewed the paragraph, made some modification and included a reference suggested. However, the estuary, which is the link between these two systems is little known, if at all (Sawakuchi et al., 2017; Ward et al., 2017). Valerio et al., (2018) collected discrete samples for $CO_2$ partial pressure all the way to the river mouth in April 2017 but does not address the Amazon River CO2 flux budget. Chen et al. (2013) studied the $CO_2$ in the world's coastal seas by evaluating the air-sea exchanges of $CO_2$ in 165 estuaries, but no data were available in the Amazon estuary, despite being arguably one with the strongest impact. Since then, Araujo et al., (2017) collected discrete DIC and total alkalinity (TA) samples at the mouth of the Pará-Tocantins River system, near the town of Belém.*

76-78: Are you referring to Argo or BCG-Argo floats? How much of the data gap in the surface ocean fCO2 was covered by the BCG-Argo floats in the open ocean? Can you put this in numbers? Do you know why there are no BCG-Argo floats in these coastal areas?

*While the statement is true for both normal and BGC-Argo floats, it is more relevant to talk about BGC-Argo floats, so we replace the sentence by: While data gaps in the open ocean have begun to narrow, partly due to advancements such as the Argo biogeochemical float program, it is not the case for biogeochemical measurement on the shelves and continental margins.*

*The number of BGC-Argo floats worldwide is limited, and most are equipped only with oxygen sensors (and occasionally nitrate). Few have pH sensors necessary for indirectly deriving $pCO_2$. Additionally, like most Argo floats, they are programmed to remain at depths greater than 1,000 m, making them unsuitable for coastal measurements, except for specific programs, like the study of the Southern Ocean's shelves. Furthermore, Brazil does not currently contribute to the BGC-Argo program, despite being well-positioned to deploy floats in its coastal waters. For the contribution of BGC-Argo floats to surface ocean $fCO_2$, we can recommend this paper in the Southern Ocean: https://agupubs.onlinelibrary.wiley.com/doi/full/10.1029/2019GB006176*

*In this paper, we agree with the other reviewer that we should insist on the fact that while Argo floats are helpful, we should aim at increasing the direct fCO2 surface measurements as they are the most accurate, especially in extreme conditions where approximating total alkalinity with salinity and temperature is highly uncertain. We therefore modified the paragraph in the paper: Continuous surface fugacity of CO2 (fCO2) measurements carried out on ships remain the most accurate way to asses CO2 fluxes and are still too sparse.*

78-81: I suggest you reorganize these sentences. There is a global decrease in the uploaded dataset of ship observations into SOCAT that is an interesting discussion to be included here. Start addressing why this might be happening and which region is more affected, then comment about the Brazilian coast. This raises an important point that just because the data is not in SOCAT it doesn't mean that there is no data in the region. This is particularly true for global south ocean.

*Thank you, we have reorganized the sentences following your suggestion: Continuous surface fugacity of $CO_2$ (fCO$_2$) measurements carried out on ships remain the most accurate way to asses $CO_2$ fluxes and are still too sparse. A notable trend in recent years is the global decline in ship-based $CO_2$ observations being added to the Surface Ocean $CO_2$ Atlas (SOCAT) database (Bakker et al., 2016), particularly since 2017 (Friedlingstein et al., 2023), mainly due to reduced fundings (Dong et al., 2024). Despite recent contributions documented in publicly available open-access data, the Brazilian continental margins remain notably under-sampled, with an acute lack of data during specific seasons, such as from August to November (Fig. 1c).*

*There has been a decline not only in the number of ship-based $CO_2$ observations uploaded to SOCAT but also in the total amount of data collected across most ocean basins. This is largely due to reduced fundings, as well as changes in merchant shipping practices and shifts in global trade dynamics. We included the reference to Dong et al, (2024) (https://agupubs.onlinelibrary.wiley.com/doi/full/10.1029/2024GL108502) that analyzed the impact of this issue on the accuracy of the CO2 uptake estimates. While research vessels*

*now contribute more data through continuous measurement systems, access to these datasets is often restricted. Additionally, some national centers maintain datasets (such as those from the Anacondas cruises), but there is no clear indication that data availability in these repositories is increasing.*

*We acknowledge the reviewer's point that data may exist in these regions outside of SOCAT. To clarify this, we have updated the manuscript to* specify 'publicly available open-access data.' *However, as far as we know, SOCAT remains the primary international reference for oceanic pCO$_2$ observations, including in the Global South.*

87: Why is the equilibrator system more accurate than membrane systems?

*Unlike membrane systems, which rely on gas diffusion through a semi-permeable barrier and can introduce biases due to response time lag and sensitivity to salinity or biofouling, equilibrator systems ensure a more complete gas exchange between water and air within a controlled volume. This allows for near-equilibrium conditions, minimizing uncertainties associated with diffusion limitations. Additionally, equilibrator systems enable regular calibration with reference gases, ensuring greater accuracy. In contrast, most membrane systems do not allow for in situ calibration as easily, making it more difficult to quantify uncertainty or correct for sensor drift (but not impossible, some of them are calibrated with reference gases). Given these advantages, equilibrators remain the preferred method for high-precision pCO$_2$ measurements in our field.*

91-94: The sentence is not completely true in the way it is written. There is the novelty of the sampling in the continuum using a sailboat. So, make sure to write this clearly, because it tends to give the idea that there is no data in these regions you refer to, which is not true. Also, it is very important to have in mind, not only for these lines but for all document, that this dataset is not filling the region data gaps, as it is just a snapshot of the conditions while the boat navigates in these particular areas. You can say that this data contributes to better understanding the area. Please, see the references suggested and provide new references that might be missing in there.

*We think there is a misunderstanding here, because in the sentence "Tara missions are unique in that they are continuous for a multi-year duration, with scientists and sailors taking turns on-board", we want to emphasize that it is the way Tara's mission are designed that is unique, and we do not intent to convey any information on data coverage. We therefore rephrased:* Tara missions have a unique design, they are continuous for a multi-year duration, with scientists and sailors taking turns on-board.

*Nevertheless, we also iterate that despite the references suggested, there is very little data made freely available for this region and this season.*

Introduction: You should focus the manuscript only in the Amazonas River Plume and NBC, as you give more attention to this area along the manuscript. Otherwise, you need to explore more and give more overview about the rest of the areas: Vitória-Trindade seamounts, Guanabara Bay, Lagoa dos Patos, and the rest of the Brazilian continental shelf.

*We do not think focusing the manuscript only on the Amazon River and River Plume is a good idea as the idea is to publish the whole dataset, that has been acquired and validated in the same way. We therefore agree with the reviewer that the rest of the areas need a more detailed overview in the introduction and added a full paragraph, including some of the references the reviewer kindly suggested.*

*The Brazilian continental shelf hosts diverse $CO_2$ flux dynamics influenced by regional oceanographic and biogeochemical processes. The ARP plays a key role in air-sea $CO_2$ exchange, with strong seasonal variability driven by river discharge, biological productivity, and salinity gradients (eg. Lefévre et al., 2010; Mu et al., 2021; Olivier et al., 2024b). In the North Brazil Current (NBC) region, upwelling and mesoscale eddies contribute to $CO_2$ flux variability, modulating carbon exchange between the ocean and atmosphere (eg. Monteiro et al., 2022; Olivier et al., 2022). Further south, the Vitória-Trindade Seamount Chain interacts with regional currents (Napolitano et al., 2021), influencing nutrient transport and biological activity that can affect $CO_2$ fluxes. The Lagoa dos Patos and Guanabara Bay are important estuarine systems where terrestrial carbon inputs, tidal mixing, and anthropogenic influences create spatially and temporally variable $CO_2$ flux patterns (Cotovicz Jr et al., 2015). Along the broader Brazilian continental shelf, complex interactions between ocean circulation, biological productivity, and local conditions shape regional carbon dynamics, making in situ observations critical for understanding these fluxes.*

95-96: Delete: "lesser studied".
*We modified the sentence.*

107: 14.000 km is a long area, therefore only the dates don't give the real visualisation of the size of each Leg. It would be also very useful if you put the km or range of lat/long of each Leg. See suggestion for figure 1 about the colours.
*Thank you for the suggestion. We have now added the latitude ranges of each leg in the table to provide a clearer sense of the spatial coverage. However, since our analysis is not structured by individual legs, we believe that adding this information to Figure 1 would unnecessarily complicate the visualization. Our focus is on the broader regional patterns rather than the segmentation by leg, and we aim to keep the figure as clear and uncluttered as possible.*

104: In lines 90-91 you say that the measurements in the South America coast were between August and December 2021. And this line is saying until November 2021. Which one is correct?
*Thank you, we modified, it is end of November.*

110: Please, correct throughout the document (including figures). The correct name of the city is "Salvador".
*Thank you, modified in the text and in the figures.*

113: Please, correct throughout the document (including figures). The correct name

of the city is "Buenos Aires", it doesn't have hyphen.
*Thank you very much, modified.*

117: Why did you choose this equilibrator system? Would be nice if you could include more details about the equilibrator system instead of just putting the reference, also with a figure/photo in figure 2.
*We selected this equilibrator system for several reasons. First, its compact size makes it more suitable for a sailboat compared to the larger GO system. Additionally, our collaboration with the research group in New Hampshire, which has successfully deployed this system in the coastal waters of the Gulf of Maine, provided an opportunity to test its performance in a new and challenging environment.*

*Regarding the request for a photograph, we believe that the schematic in Figure 2 provides a clearer and more rigorous representation of the system's setup and functioning. A photograph, while useful in some contexts, may introduce visual complexities that do not necessarily enhance the technical understanding of the system. However we included a brief description of the system's physical setup in the text to complement the schematic.*
*The fCO2 system uses a shower spray air–sea equilibrator of 2.5 L as described by Dickson, (2007) and used by Vandemark et al., (2011). Water is sprayed or trickled inside a chamber, creating a large surface area for rapid equilibration with the headspace air. A closed loop of air flows through the equilibrator where the air-water exchanges happen, the equilibrated air is drawn at 100 mL/min through tubing containing a Nafion selectively permeable membrane with a counterflowing stream of dry nitrogen to remove water vapor from the sample gas stream. It is then sent to a non-dispersive infrared CO2 analyzer, a LICOR LI-840A.*

119-120: "It is able to capture the fine scale variability of oceanic fCO2 by responding quickly to fCO2 changes in seawater" This seems very vague, please, include reference or rephrase.
*The next sentence of the manuscript provides both the reference and the values to justify this statement: The exchange time for the water in the equilibrator is between 30 and 45 seconds, depending on flow rate (Pierrot et al., 2009).*

123: How many square meters? Use precise numbers.
*2.5 m$^2$, modified in the text.*

Figure 2: You could include a picture of the equilibrator system. Please, include where SBE 38 is in the scheme.
*Thank you, we included the SBE38 in the schematic. Regarding the request for a photograph, see comment above.*

[Figure]

*Figure 2: Revised version of Figure 2*

142: How the equilibrator air was dried? Please, include as much as information you have for this methodology as it can be replicated in future years. Including the sampling rates for STD gases, atmospheric and ocean.

*Thank you very much, we added: "the equilibrated air is then dried using a gravity water trap and Nafion tube before being sent to the $CO_2$ analyzer", to indicate how the air is dried. We added the information regarding the sampling rates of STD gases and atmospheric CO2:*

*During the first week, to test the system, a complete set of standards and atmospheric cycle was measured for 15 minutes every hour.*

144: I couldn't find the Annex.

*The information is in the Annex of Pierrot et al., 2009. To improve clarity, we modified in the text: "It detects the molar fraction of CO2 (xCO2) in dry air by infrared detection, from which fCO2 is computed following Henry's law (Pierrot et al., 2009, detailled in their Annex)".*

149: Please, describe how these valves are controlled?

*They are electro-valves. We added the information to the manuscript: "Through a system of electro-valves".*

151: Why were these two standard gases chosen? There is a reference that suggests that or is something you are suggesting for the first time? As the mission would measure the river-ocean continuum, why didn't you choose a STD gas with a higher concentration to include the values between 0 and the maximum value found in the river, as recommended in SOCAT Cookbook (2018)?

*Thank you for this insightful question. The selection of these two standard gases was carefully made to balance multiple objectives. As you correctly pointed out, SOCAT recommends using standard gases that bracket the expected values. However, it is equally important that the standards remain as close as possible to the measured values to minimize calibration uncertainty.*

*Using a higher standard gas (e.g., 1000 or 2000 µatm) would have significantly reduced the accuracy of oceanic measurements. We chose 0 and 500 µatm because they effectively bracket the range of oceanic fCO$_2$ values in this highly variable environment, encompassing*

*most of the observed data except for the river. Since this system was designed for long-term deployment in a predominantly oceanic setting, these values were deemed the best compromise.*

*To address this, we have now clarified in the manuscript the rationale behind our choice and explicitly stated that measurements beyond this range (e.g., in the river) have greater uncertainty due to the lack of higher standard gases.*

*In the text: "The two 20 L reference gases tanks of 0 ppm and 502.3 ppm are stored on the front deck. These values were chosen because they effectively bracket the range of oceanic $fCO_2$ values in this highly variable environment, encompassing most of the observed data except in the river. As a result, $fCO_2$ values above 500 µatm are more uncertain and should be interpreted with caution".*

154-155: "It is recommended to measure a complete set of standards every 3 hours."-Include reference. Can you explain more about these changes, in which leg it was made, for example.

*Thank you for the comment, we included the reference to Pierrot et al., (2009) and added the date at the time the change was made: the measurement of standards was increased to every 6 hours (on 31/08), then every 12 hours (on 02/09) to save the reference gases*

163-164: How regularly the equilibrator was cleaned? There were other methods to avoid mud in the system?

*The equilibrator was cleaned at each stopover and each time we exited a major river. We added this information to the manuscript.*

*The system also has the advantage to be able to work in turbid environment. The equilibrator was cleaned at each stopover, and each time the ship exited a major river (so 7 times in total) to avoid the buildup of mud.*

168: Why did you proceed with this STD gas if it wasn't in the range reported by the supplier? Also, did you try to measure in a different analyser to see if the problem was the gas or your LICOR?

*We appreciate the reviewer's suggestions and acknowledge that further testing of the standard gas would have been ideal. However, due to logistical constraints and limited time in port after receiving the gas in Martinique, additional checks were not feasible. To independently assess the gas, we relied on a comparison with the site in Barbados, which exhibits minimal variability when the wind comes from the sea. This approach provides a reliable calibration reference for the standard gas. The LICOR didn't have any issue with the other standard gas (0) and when back in the laboratory after the Tara mission.*

Figure 3: Please, increase the legend font. I suggest a scatter plot to make the comparison clearer and more realistic. Also, a statistical metric to support the relation between the datasets.

*Thank you, we increased the fonts. We think the scatter plot suggestion might be for figure 4 instead of 3? We included a scatter plot in the new version of Figure 4.*

180: There is an inconsistency between the dates in the figure 3 (18-19/08/2021) and the main text (19-20/08/2021). My concerns about this correction is: i) it used just 7h of measurements, in just one point in the early stage of the campaign. Is this representative? Do you think it is possible to compare to another dataset or increase the calibration curve with the RPB measurements?

*Thank you for catching the inconsistency; we have corrected our mistake. We acknowledge that the comparison period is relatively short, but there is no reason to expect that the standard gas values would have drifted over time. Additionally, given the conditions, there is no strong indication that a significant spatial gradient in atmospheric $fCO_2$ existed between the ship and the land station. Furthermore, we examined the atmospheric $fCO_2$ values to ensure the robustness of our assessment.*

185-189: Which method? Compared the values and decreased one from the other? This is not a reliable calibration method for a long dataset as presented and with all unstable conditions for the gas cylinder. Please, provide a more reliable method that uses a significance range, calibration factor, or something that ensures your data is correctly calibrated and it is possible to replicate your calibration in other parts of the campaign.

*Apologies for the confusion; we have replaced 'method' with 'approach' to better reflect our intent. This approach was not used to calibrate the entire dataset but rather to determine the correct value of the standard gas. The rest of the calibration follows standard procedures applied on research vessels. There is no indication that the gas cylinder was unstable or that its properties changed over time, as such stability is generally expected and routinely assumed in similar shipboard measurements.*

193: Even though Metzl et al., 2024 provide the synthesis of SNAPO-CO2-v1 dataset, you need to be able to provide a brief explanation of the TA and DIC methodology used in your campaign, especially using these data to validate you fCO2 dataset.
You also mentioned 17 samples for the surface, but you present only 13 in figure 4. Please, be clear in the text how many samples you used.

*There were 17 samples collected, but four were either not analyzed correctly (in the Amazon river) or were not collected with simultaneous pCO2. To clear the confusion, we added in the description of Figure 4: The fCO2 system was measuring the standards during stations 36abc and 39, these stations are therefore not represented in (d). We added information on the sampling methodology for DIC and TA: Samples were drawn from the rosette into 0.5 L borosilicate glass bottles, ensuring minimal air contamination, and immediately poisoned with 400 µL of mercuric chloride ($HgCl_2$) to prevent biological alteration. TA was measured using open-cell titration with a hydrochloric acid titrant, while DIC was analyzed using acidification followed by $CO_2$ extraction and detection via infrared or coulometric*

*methods. Quality control was ensured through calibration with certified reference materials to maintain an accuracy of ±4 μmol kg⁻¹ (Metzl et al., 2024)*

195: Provide which version of CO2SYS you used and the reference.

*Thank you, we used CO2SYS v3.1 and included both the version and the reference in the manuscript: The $fCO_2$ is computed from the near-surface ocean DIC and TA using the CO2SYS v3.1 software (Sharp et al., 2020) to compare with the continuous fCO2 measurements (Fig. 4).*

Figure 4: This figure gives a wrong perception of distance between the discrete samples. The figure could be sliced by leg to better visualize. In the current way it is not possible to address the values very easily. I also suggest a table where one column is the fCO2 calculated by the CO2SYS and the other by the equilibrator.

*Thank you for your comment. We heavily modified Figure 4 to also integrate the useful suggestions from the other reviewer. We hope this second version, with different ranges of fCO2 values and a scatter plot addresses your request to be able to better compare the fCO2 calculated and measured.*

[Figure]

*Figure 3: Revised version of Figure 4*

205: "the continuous fCO2 compares very well to the one computed from the samples, especially after 26 September" – It is not possible to see this in the figure 4, and no statistical method was applied on the dataset to prove or support this sentence. Again, this is a long area with high variability, a comparison as presented in fig 4 is a weak assumption that your data has the required accuracy, especially for the river areas where you don't have neither discrete samples or the STD gas.

*The comparison is explicitly and statistically reported on line 210. We acknowledge that in the river area, the accuracy is lower, partly due to the limitations of the DIC/TA samples collected. Unfortunately, the analysis method used did not allow us to retrieve reliable DIC and TA values, adding further uncertainty in estimating pCO2 in these waters. We have now explicitly stated this in the manuscript (see answer to next comment). While the number of discrete samples remains limited, and the uncertainty from atmospheric xCO2 comparison*

*should be considered, we have no indication that the data accuracy is lower than what is reported. Additionally, we have modified Figure 4 to enhance the visibility of the comparison.*

217: Please, check if your data matches the ones in the suggested references.
*Thank you, in answer to this and the comment before, we modified the paragraph accordingly:*
*Overall, the mean difference remains around 2 µatm, providing a reasonable estimate of the dataset's uncertainty. In the river, where fCO₂ values fall outside the range of the standard gas used and no discrete samples are available for direct comparison, the uncertainty is likely higher. However, the values obtained align with expected ranges for this part of the river, based on discrete samples collected in April 2017 by Valerio et al. (2022), despite differences in season and year.*

226: I strongly recommend you submit to SOCAT 2025 only the data for Amazonas River plume, as you didn't provide a reliable calibration and validation to the other parts of the dataset, especially in the rivers.
*We submit the entire dataset, as it follows the same validation and calibration procedures applied to other ship-based fCO₂ observations included in SOCAT. While we acknowledge that uncertainty is higher for the elevated fCO₂ values observed in the estuary and near Macapá and Belém, we explicitly state this in the manuscript. SOCAT includes datasets with varying levels of uncertainty, provided that these uncertainties are well-documented, which is the case here.*

Figure 5: Please increase axes and colour bar font. I appreciate the consistency in the maps, however, it is not very easy to see the dataset variation in this. I suggest dividing by the legs. Please, include geopolitical map and include names (especially the ones you use in the text).
*Thank you for your suggestion. We increased axes, and changed the projection to increase the focus on the dataset and remove as much white as possible. We added the names of the main cities, and as commented previously we refrain from adding the borders to comply with the ESSD guidelines.*

[Figure]

*Figure 4: Revised version of Figure 5.*

238: 36 is considered salty waters.
*Thank-you. We replaced 'saline' by 'salty'.*

238: What does "recent ARP" mean in this context?
*Thank-you. We rephrased by: The schooner then crosses the salty (36) water of the NBC retroflection, before sampling the river plume that has been recently transported from the Amazon estuary.*

248: This can be due to rain in the land, which increases the river discharge, please check this and the references.
*We have reviewed rainfall data for the period in question and found no significant rainfall event near the estuary. Instead, we suggest that changes in wind patterns, particularly wind direction, may have played a more substantial role. We have previously discussed this in greater detail in Reverdin et al. 2021.*

249: Change "maritime" to "marine". Please revise this in all document.
*Thank you, done.*

Figure 6: It gives more the idea of the places however the fCO2 axes need to show with more numbers.
*Thank you, we increased the number of ticks for the fCO2 on Figure 6.*

267: Please check the suggested references.
*Thank you, we included the reference to Napolitano suggested.*

268: The correct name of the city is Rio de Janeiro.
*Thank you, modified.*

270: Which is considered low salinities in Santos?
*Thank you, we indeed modified low by lower. The salinity is 34.8, which is still quite salty. In the manuscript: It shows strong fCO2 variability, with low values associated to the lower salinities (34.8) close to Santos.*

Figure 7: Please include the name of the cities and countries borders.
*Thank you very much, we added the name of the main cities. For the borders, we follow the ESSD guidelines.*

[Figure]

*Figure 5: Revised version of Figure 7.*

280: Please provide Lat/Lon of the center of the plume first here, and indicate in the figure 7.

*Thank-you. Following your comment, we realize that 'center of the plume' might not be the best wording, and rephrased the sentence as: The minimum observed $fCO_2$ in the Amazon River plume is of 65 µatm (4.5°N/50.77°W), whereas outside of the plume, in the NBC, $fCO_2$ is around 420 µatm.*

282: "towards the Amazon" what? This seems incomplete.

*Thank you for noticing the missing word, we complete with "towards the Amazon estuary".*

Table 1 is not very informative as you could include this information in the text. If you follow the suggestion to focus the manuscript only in these areas, it would be interesting to see the difference of the mean fCO2 for these regions which you could include in this table.

*We acknowledge that Table 1 may not be particularly informative, and therefore we removed it. Its purpose was to ensure full reproducibility of the figures presented in the manuscript, so we adapted Figure 9 to include the information that was in Table 1. As this is primarily a data paper rather than a scientific analysis, we provide insights into potential uses of the data through the figures, while leaving the scientific interpretation to the users of the data. We do agree that information on fCO2 would be valuable. However, we leave this level of analysis to the data users who may wish to conduct more targeted studies on the subject.*

[Figure]

*Figure 6: Revised version of Figure 9.*

291: "follow well the relationship reported in Lefèvre et al., (2010)". Please provide a statistical metric that supports this, especially because the river region (brown) doesn't look like it fits well.

*We completely agree with the reviewer, that is why the sentence states that the data follow well the relationship "From the NBC to the core of the ARP (6°N to 4°N)", so not the river region in brown. That is also why the next sentence is "However, when salinity and fCO₂ increase locally from 4°N to 2°N, they move away from the NL linear relationship". We added the colors in parentheses to make it clearer.*

292: What does "NL" mean?

*Sorry, we didn't include the definition of the acronym, it stands for Lefèvre et al. (2010) fCO2/SSS relationship. We now explicitly mention it in the manuscript: the fCO₂/SSS measurements follow well the relationship reported in Lefèvre et al., (2010, then NL).*

Discussion: Needs to be revised after including the key references and data validation metrics, and study area reduction to Amazonas River plume.

*As previously mentioned, we do not agree with limiting the dataset to a specific subset. We have included the necessary validation metrics, which are critical for an ESSD paper, as well as additional references as suggested.*

324: Tropical band is from 20°N to 20°S, where the tropics are. Perhaps you mean the equatorial area?

*Correct, thank you very much, it has been modified.*

332: "changes in biogeochemical and biological properties." You didn't have biological data, so please include a reference for this sentence.

*There were biological measurements during the cruise, but they haven't been published yet and we don't comment on them, and thus removed 'biological'.*

335: Provide the agreement with salinity.

*Thank you, this paragraph has been rephrased.*

343: "Nevertheless, while the large-scale variability of the fCO2 reflects the latitudinal temperature gradient" – include references that support your findings. See suggested references.

*Done*

345-346: "Other river discharges reach the south Atlantic, such as the one of the Rio de la Plata." – Include references.

*Done*

l.346-347: "These waters spread on the shelf and generate variability in salinity, suspended sediments and biological activity." – Include references.

*Done*

352: Update reference.
354: Update reference.
361: Update reference.
*If that is ok, we kept the original reference because it refers to the first time these processes where identified, and added more recent ones.*

361 – 362: "The source to sink transition is mainly driven by the switch from a respiration dominated system to a photosynthetic one." – Include references.
*Done*

362-363: "Several factors impact the suspension of sediments in the water column and the development of phytoplankton, such as the bathymetry, winds and the tides." – include references. Include the rainy season, ITCZ position and, El Niño and La Niña occurrence impact.
*We modified by: Several factors impact the suspension of sediments in the water column and the development of phytoplankton, such as the bathymetry, winds, intensity of the outflow (that can be influenced by large-scale climatic modes) and the tides (Gomes et al., 2021).*

365: Update reference.
*Done*

388: "There are very little previously acquired data in the region that can be used for comparison." - Check references.
*Unfortunately, there are little directly relevant data. There are indirect observations, but almost none of fCO$_2$ during this particular season, and that cover the Amazon estuary-ocean continuum. We rewrote this sentence to be more specific. As mentioned, the SOCAT 3-month data map presented still is a rather complete map on what was collected during this season. Thus, actually, although there are some data further north and south that are not in the SOCAT data base (such as from Anacondas cruise), there is no other pCO2 data within 1°S and 6°N in the published literature (except for the river, but as mentioned, this is not the focus of the study, and except near Belem and Para River, but missing most of the Amazon estuary).*

393: Delete "if they would exist." Check references first.
*We agree and modified the sentence by: it would be more accurate to cross-compare with fCO$_2$ measurements conducted in the same region at the same time as recommended by SOCAT. In the quality-checking process of SOCAT, the fCO2 measurements need to be conducted in the same region but also at the same time (same year, same month) to be able to qualify as a flag A. Despite the interesting references provided, no fCO2 measurement was conducted in the region in September-October 2021.*

408: Delete "which had never been observed before." - Check references first.

*Thank you for the references provided, we checked them and we can maintain that this statement is correct for the period. We therefore modified to: "which had never been observed before in this season".*

408-409: Change "for the first time a sailboat equipped with a fCO2 system is sampling the river-ocean continuum". Otherwise, this is an overinterpretation, especially because your river-ocean continuum data is not reliable.

*Thank you for your comment. We respectfully maintain that our statement is accurate. We originally wrote: "For the first time, a schooner equipped with an fCO$_2$ equilibrator system measured fCO$_2$ along the eastern coasts of South America," which remains factually correct, as no other sailboat has been equipped with such a system for this purpose.*

*We acknowledge the reviewer's concern regarding data reliability in riverine conditions. However, we would like to clarify that the dataset collected in oceanic conditions meets the same reliability standards as other fCO$_2$ datasets. We agree that data quality is more uncertain in riverine conditions, but this primarily applies to a limited portion of the dataset and does not affect the measurements taken from Belém to Uruguay or from Martinique to the Amazon River.*

416: "filling part of the data gap in the coastal regions of the South Atlantic Ocean", this is an over-interpretation, delete or rephrase.

*We rephrased by: providing data in the data-poor region of the coastal regions of the South Atlantic Ocean.*

420: Delete. The only mention of Guanabara Bay is in the conclusion, and it is not easy to find the data of these regions in your result.

*Agreed, we remove the mention to the Guanabara Bay.*

---

## Referee Report (RR1)

**2nd Review for the manuscript: "Exploring the CO₂ Fugacity along the East Coast of South America aboard the Schooner Tara" by Olivier et al.**

**Time devoted to this review**: 5 hours.

**Reviewer**:
Marcos Fontela (IIM-CSIC)

I'm happy to see that the authors put in significant effort to evaluate all reviewer's suggestions, not only mine. I'm feel overall satisfied with the changes and the responses. I honestly think that the revised version is improved compared to the original one. Data product is clearer now. That was my main concern considering the journal we are dealing with. Figures are also improved.

I would like to see this article accepted after minor edits and technical corrections are done.

In the last lecture of the manuscript I only did the following annotations:

This citation could be added to lines 57 and/or 80 and/or 86 and/or 90 https://bg.copernicus.org/articles/7/1587/2010/

Other useful cites: https://link.springer.com/article/10.1007/s00267-015-0630-x

Sentence ending in line 95 "sparse (CITATION?)." I miss a citation here.

Table 1. Buenos Aires arrival date is missing. From August (08) to November (11) there are 3-months, not 4 as you say later in the conclusions.

Line 648 (tracked version): The temporal resolution of "fCO2 measurements every minute" is mentioned here for the first time. This should be introduced earlier in the methods or results section, not in the conclusions.

Line 649 (tracked version): The dataset spans a four-month period but is not continuous. The conclusions should explicitly state the total number of days and hours with valid observations to clarify the temporal coverage.

Line 650 (tracked version): The inclusion of the standard deviation here is not informative, as it is inflated by the high values from the Amazon River plume.

Line 656 (tracked version): I would replace "*heart of the plume*" with "*core of the plume*" (?) to avoid physiological metaphors.

---

## Author Response (AR2)

**Response to reviewers & editor**

**Editor**

After thorough review of the manuscript and consideration of all reviewer comments, I recommend publication of this dataset subject to minor revisions as suggested by the reviewers. The information presented constitutes a valuable contribution to the field that merits publication, even though some concerns have been raised about the calibration procedures. The dataset provides interesting and potentially useful information that fills an existing knowledge gap.

I suggest including, around section 2.3, some sentences to acknowledge and document in the article the concerns of reviewer #2:

"I still have concerns about the calibration approach as no statistical test or method was used for that. As exposed by the authors, they just adjusted the STD gas value to the result it should give in comparison to the Ragged Point, Barbados (RPB) station. They have insufficient discrete samples to compare with the underway measurements, which also don't match on the AROC, the site with greater variability. The scatter plot helps to see the tendency of an agreement between measurements (discrete and continuous) in the oceanic part; however, it needs to take into account that it is just 5 points for 14,000 km, which is insignificant for a region with high variability."

Even though it is not possible to solve the issue due to lack of data, it is worth mentioning this potential weakness.

Once the authors have addressed the minor revisions outlined in the reviewer reports, the dataset will be accepted for publication in ESSD

*Dear Editor, thank you very much for support and guidance throughout the review process. We appreciate your comments, and constructive suggestions, which have been very helpful in improving the manuscript.*

*We have addressed all the minor revisions suggested by the reviewers, as detailed below. In particular, we have added several sentences to Section 2.3 to transparently acknowledge the concerns raised by Reviewer 2 regarding the calibration approach and the limited number of discrete samples. Our aim was to clearly communicate the limitations of the dataset and ensure users are appropriately informed when interpreting the data. We added: As no simultaneous dataset can be used to cross-quality check the data, the agreement tendency between fCO₂ estimated from 5 samples and the continuous fCO₂ measurements is important, and is used here to validate the data. However, the number of samples, particularly in the open ocean, is very limited relative to the distance covered, which limits the statistical robustness of the validation of both the dataset and more importantly of the calibration approach. Users should therefore interpret the data, especially in coastal and river-influenced regions, with appropriate caution.*

**Reviewer 1**

I'm happy to see that the authors put in significant effort to evaluate all reviewer's suggestions, not only mine. I'm feel overall satisfied with the changes and the responses. I honestly think that the revised version is improved compared to the original one. Data product is clearer now. That was my main concern considering the journal we are dealing with. Figures are also improved. I would like to see this article accepted after minor edits and technical corrections are done.

In the last lecture of the manuscript, I only did the following annotations:

This citation could be added to lines 57 and/or 80 and/or 86 and/or 90
https://bg.copernicus.org/articles/7/1587/2010/
Other useful cites: https://link.springer.com/article/10.1007/s00267-015-0630-x
*Thank you pointing us towards these two interesting papers. The first reference has been included line 55 and the second one line 81.*

Sentence ending in line 95 "sparse (CITATION?)." I miss a citation here.
*Thank you we included a citation toward the global carbon budget 2024, that shows the unexplained and unresolved difference between the observed and model CO2 sink (one of the possible explanations being the lack of surface ocean CO2 data).*

Table 1. Buenos Aires arrival date is missing. From August (08) to November (11) there are 3-months, not 4 as you say later in the conclusions.
*Thank you for the valid remark, we didn't include the data of arrival to Buenos Aires because the dataset stops before. Furthermore, it doesn't stop on November 11$^{th}$ but at the limit of the Uruguayan EEZ on November 25$^{th}$, and the end-date is now modified.*

Line 648 (tracked version): The temporal resolution of "fCO2 measurements every minute" is mentioned here for the first time. This should be introduced earlier in the methods or results section, not in the conclusions.
*Thank you for pointing it out, it is now included in the methods, section 2.5: "Following the recommendations of Pierrot et al. (2009) and of SOCAT, the dataset provides for each location and time step the measured data: molar fraction of CO2 in the equilibrator (xCO2eq), sea surface salinity (SSS), temperatures (SST and Teq), and pressure (Patm), the calculated variables (pCO2sw, fCO2sw), averaged over one minute."*

Line 649 (tracked version): The dataset spans a four-month period but is not continuous. The conclusions should explicitly state the total number of days and hours with valid observations to clarify the temporal coverage.
*We agree, the dataset spans over 3.5 months and we collected 45 days and 8 hours of valid fCO$_2$ coverage. In the text, we modified the sentence by: this dataset of 65,000 measurements spanning over almost 4 month (from August to the end of November 2021, for a total of 45 days and 8 hours of valid fCO$_2$ data) shows large fCO$_2$ variability.*

Line 650 (tracked version): The inclusion of the standard deviation here is not informative, as it is inflated by the high values from the Amazon River plume.
*Thank you, we agree and removed the standard deviation.*

Line 656 (tracked version): I would replace "heart of the plume" with "core of the plume" (?) to avoid physiological metaphors.
*We agree, and replaced heart of the plume by core of the plume.*

**Reviewer 2**

Dear,
I would like to acknowledge the authors for the responses to the suggestions and concerns that were raised and the good job they did to improve the manuscript after the major reviews. The manuscript is better written and clear, the figures are more comprehensive and contribute more to the understanding of the dataset now.
Due to the changes, I agree with the authors that the dataset should be submitted as whole, and not only the Amazonas River-Ocean Continuum (AROC) part as suggested previously.
I still have concerns about the calibration approach as no statistical test or method was used for that. As exposed by the authors they just adjusted the STD gas value to the result it should give in comparison to the Ragged Point, Barbados (RPB) station. They have insufficient discrete samples to compare with the underway measurements, which also don't match on the AROC, the site with greater variability. The scatter plot helps to see the tendency of an agreement between measurements (discrete and continuous) in the oceanic part, however it needs to take in account that is just 5 points for 14.000 km, which is insignificant for a region with high variability.

*Thank you for raising this important point. We fully understand the reviewer's concerns and agree that the limited number of discrete samples, particularly over such a long and variable transect, restricts the robustness of any statistical validation of our calibration approach. Given the constraints of the dataset, we chose what we considered the most appropriate method, but we recognize its limitations.*
*To address this, we have revised Section 2.3 (Validation) to explicitly acknowledge these limitations and emphasize the need for caution when interpreting the data—especially in dynamic or river-influenced regions. Our aim is to be fully transparent with the users about the calibration choices and the associated uncertainties. We added:*
*As no simultaneous dataset can be used to cross-quality check the data, the agreement tendency between $fCO_2$ estimated from 6 samples and the continuous $fCO_2$ measurements is important, and is used here to validate the data. However, the number of samples, particularly in the open ocean, is very limited relative to the distance covered, which limits the statistical robustness of the validation of both the dataset and more importantly of the calibration approach. Users should therefore interpret the data, especially in coastal and river-influenced regions, with appropriate caution.*

I also have a few minor comments on some sentences in the manuscript that I included below.

Regarding terminology, the authors use the word "significant" in the manuscript but they don't have any metric to support it. Therefore, I suggest they change it through the whole document, for example in:

210-215: In "Tara crossed highly variable regions during its voyage, supporting our confidence that the uncertainty on the dataset due to this span value is not significantly impacting the results".

*Thank you for the comment. We have* replaced the word "significant" throughout the manuscript with more appropriate alternatives, except in the introduction where the phrase "significant carbon sink" is supported by previous studies. In particular, around lines 210–215, we also took the opportunity to highlight the limitations and uncertainties of the calibration approach, in line with the concerns raised in previous comments.

*Tara crossed highly variable regions during its voyage, supporting our confidence that the uncertainty on the dataset associated with this span value has limited influence on the overall results, but should nevertheless be take into account when analyzing the dataset.*

410-415: Nevertheless, we observe significant variability of fCO2 even if the salinity does not change anymore.

*In this example, we* removed the word significant

250-255: In "As no simultaneous dataset can be used to cross-quality check the data, the good agreement between fCO2 estimated from the samples and the continuous fCO2 measurements is quite important, and the mean differences are in the range of uncertainties related to inferring fCO2 from the DIC and TA measurements."

This is not possible to say as in the manuscript there are only 5 samples to compare (figure 4). My suggestion is to delete or rephrase - suggestion "As no simultaneous dataset can be used to cross-quality check the data, the suggested agreement/agreement tendency between fCO2 estimated from 5 samples and the continuous fCO2 measurements is important, and is used here to validate the findings as the mean differences are in the range of uncertainties related to inferring fCO2 from the DIC and TA measurements."

*We agree, thank you for the nice reformulation, we just modified 5 by 6, it is* now included in the manuscript.

405-410: In "The zone of lowest SSC is between the isobaths 10 m and 20 m, associated to a diatom bloom in 1983 (Curtin and Legeckis, 1986; DeMaster et al., 1986). It is also the region where we observe the transition from a source to a sink of CO2 (Fig. 9)."

I don't understand the relevance of mentioning a bloom from 40 years ago. As a reader, it gave the impression that it is a recurrent event that justifies the authors finding of the transition of behaviour (source to sink).

Now I know that the authors want to compare their results with the first literatures and descriptions in the region, but for this sentence my suggestions are to see if there is an

updated literature about the suspended sediment characteristics of these zones, or some literature that supports a recurrent bloom in this area. If none is available, simply rephrase. Suggestion: "The zone of lowest SSC found by Curtin and Legeckis, (1986); DeMaster et al., (1986) is between the isobaths 10 m and 20 m, which it is also the region where we observe the transition from a source to a sink of CO2 (Fig. 9)."

*Thank you for this perspective. We understand what the reviewer means, and indeed is was not in our intention to use a singular event to justify a permanent behavior. We modified the manuscript according to the sentence suggested.*

460-465: In "Despite sampling most of the American coastline along the South Atlantic Ocean, it represents only a small fraction of the world's coastlines"
This sentence plays against the relevance of the manuscript. If this is just a small fraction of the global coastlines why is important to study it then? I suggest deleting it.
*We wanted to highlight the need to increase the data coverage along the coast, but we do not want to deserve our manuscript, so we deleted the sentence as suggested.*